# READ THE ROOM: VIDEO SOCIAL REASONING WITH MENTAL-PHYSICAL CAUSAL CHAINS

**Lixing Niu** [1,2]   **Jiapeng Li** [3]   **Xingping Yu** [4,2]   **Xinyi Dong** [5]   **Shu Wang** [6]
**Ruining Feng** [7]   **Bo Wu** [8]   **Ping Wei** [3]   **Yisen Wang** [1]   **Lifeng Fan** [2 †]

† Corresponding author.
[1] School of Intelligence Science and Technology, Peking University
[2] State Key Laboratory of General Artificial Intelligence, BIGAI
[3] State Key Laboratory of Human-Machine Hybrid Augmented Intelligence,
   Institute of Artificial Intelligence and Robotics, Xi'an Jiaotong University
[4] School of Psychological and Cognitive Sciences, Peking University
[5] Yuanpei College, Peking University   [6] University of California, Los Angeles
[7] Department of Automation, Tsinghua University   [8] MIT-IBM Watson AI Lab
`lxniu@stu.pku.edu.cn, lifengfan@bigai.ai`

## ABSTRACT

"Read the room", or the ability to infer others' mental states from subtle social cues, is a hallmark of human social intelligence, but remains a major challenge for current AI systems. Existing social reasoning datasets are limited in complexity, scale, and coverage of mental states, falling short of the rich causal dynamics found in real-life interactions. In this work, we introduce $R^3$-Bench, an evaluation benchmark with fine-grained annotations of belief, intent, desire, emotion, and their causal chains in complex scenarios. Furthermore, we introduce $R^3$-FDT, a large-scale training set generated through a novel automated pipeline with the same chain structure. We conduct a comprehensive evaluation of state-of-the-art (SOTA) large vision-language models (LVLMs) on $R^3$-Bench, revealing substantial deficiencies in consistent multi-step social reasoning. We also fine-tune a 7B model on $R^3$-FDT, achieving notable improvements across multiple relevant benchmarks. Our contributions are three-fold: (i) a novel benchmark with richly annotated, multi-step causal reasoning data; (ii) systematic evidence that SOTA LVLMs fall far short of human-level reasoning; (iii) a scalable training dataset that significantly enhances social reasoning performance. The datasets and codes are available at: `https://github.com/LiXingNiu/Read-the-Room.git`.

## 1 INTRODUCTION

"Read the room" requires employing Theory of Mind (ToM) (Premack & Woodruff, 1978) to read others' minds and perform social reasoning with subtle cues; it represents higher-level social intelligence, and plays a key role in helping people navigate social scenarios smoothly. Humans are innate with the ability to perceive huge hidden information from very simple cues (Heider & Simmel, 1944; Shu et al., 2018; Zhu et al., 2020; Fan et al., 2022); however, it remains a great challenge for current AI. As shown in Figure 1, the visible physical world is only the tip of the iceberg; beneath it lies a vast and often invisible mental world. In just a few seconds, people perceive layers of causally linked mental states: who is aware of what, who is hiding what, and how others respond. These interpretations rely not only on observable actions but also on unspoken norms and contextual reasoning. Effective social reasoning thus involves (i) detecting subtle behavioral cues, (ii) estimating diverse evolving mental states, and (iii) identifying causal chains connecting physical and mental worlds over time.

Large language models (LLMs) have recently demonstrated strong reasoning abilities across various domains (Brown, 2020; Wei et al., 2022; Kojima et al., 2022; Bubeck et al., 2023; Wang et al., 2024c). However, they still struggle with complex reasoning tasks such as long-term planning and scientific problem solving (Srivastava et al., 2022; Wang et al., 2024d; Mirzadeh et al., 2024; Glazer et al., 2024). Social reasoning—a crucial subset of complex reasoning—also remains challenging for

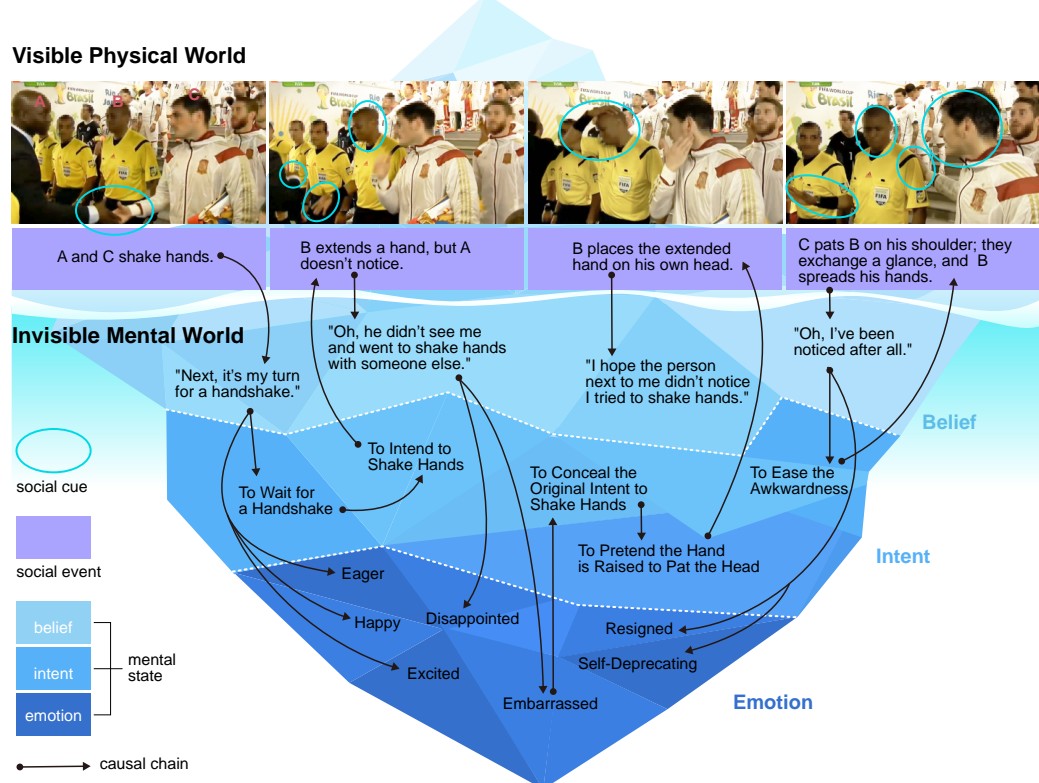

Figure 1: The visible physical world we live in is just the tip of the iceberg compared to the vast, invisible mental world behind it (Zhu et al., 2020). In this example, a brief social interaction reveals complex and dynamic mental activities: B extends his hand to shake with A, but A fails to notice. B then pretends his outstretched hand was meant to touch his head, concealing his embarrassment. C, however, sees through B's mental state and pats him on the shoulder to offer comfort. In response, B shrugs and gestures self-deprecatingly to ease the awkwardness. Social reasoning is a critical aspect of social intelligence. Yet in long-term, dynamic interactions, capturing subtle cues, recognizing social events, estimating mental states, and identifying causal chains becomes increasingly difficult, making social reasoning even more intricate.

LLMs (Shapira et al., 2023; He et al., 2023; Gu et al., 2024; Wang et al., 2024a). Critically, modeling social cognition requires visual signals to infer mental states that language alone cannot capture.

To address this, large vision-language models (LVLMs) (Liu et al., 2024; Zhang et al., 2023; Lin et al., 2023; Team et al., 2023; 2024; Hurst et al., 2024) have emerged, enabling multimodal understanding. Yet, current benchmarks provide limited evaluation of (i) diverse mental state estimation and (ii) the consistency and depth of social reasoning in complex interactions. Moreover, there remains no large-scale video dataset covering multiple mental states and their causal relationships, hindering further development in this domain. As illustrated in Table 1, current video datasets suffer from significant limitations. MVBench (Li et al., 2024), MMBench-Video (Fang et al., 2024) and Video-MME (Fu et al., 2024) primarily focus on factual or visual comprehension, offering limited support for deep mental state reasoning and causal inference. MMToM-QA (Jin et al., 2024), MELD (Poria et al., 2018) and IntentQA (Li et al., 2023) target a few types of mental states and lack multi-step reasoning. Social-IQ (Zadeh et al., 2019) and Social-IQ 2.0 (Wilf et al., 2023) incorporate social contexts but do not model fine-grained causal chains or assess reasoning across linked events. Also, the training sets of Social IQ and Social IQ 2.0, which offer the broadest (though still limited) coverage, contain merely 1k videos and 6k+ questions, respectively. It is insufficient to support the training of a foundation model capable of grasping the core, diversity, and complexity of social reasoning.

Therefore, we introduce a fine-grained evaluation benchmark named R$^3$-Bench (**R**ead the **R**oom **R**easoning **Bench**mark) and a large-scale training set named R$^3$-FDT (**R**ead the **R**oom **R**easoning

| Datasets | Real-World | Mental State | | | | Causality | CC | Training Set | | Test Set | |
|---|---|---|---|---|---|---|---|---|---|---|---|
| | | Belief | Intent | Desire | Emotion | | | # Video | # QA | # Video | # QA |
| NExT-QA (Xiao et al., 2021) | ✓ | ✗ | ✓ | ✗ | ✗ | ✓ | ✗ | 3.9k | 34k | 1k | 8.5k |
| MVBench (Li et al., 2024) | ✓ | ✗ | ✗ | ✗ | ✗ | ✗ | ✗ | - | - | 3.6k | 4k |
| MMBench-Video (Fang et al., 2024) | ✓ | ✗ | ✗ | ✗ | ✓ | ✓ | ✗ | - | - | 0.6k | 2k |
| Video-MME (Fu et al., 2024) | ✓ | ✗ | ✗ | ✗ | ✗ | ✗ | ✗ | - | - | 0.9k | 2.7k |
| MMToM-QA (Jin et al., 2024) | ✗ | ✓ | ✓ | ✗ | ✗ | ✗ | ✗ | - | - | 0.1k | 0.6k |
| MELD (Poria et al., 2018) | ✓ | ✗ | ✗ | ✗ | ✓ | ✗ | ✗ | 10k | - | 2.6k | - |
| IntentQA (Li et al., 2023) | ✓ | ✗ | ✓ | ✗ | ✗ | ✓ | ✗ | 3.2k | 12k | 0.6k | 2.1k |
| CausalChaos (Parmar et al., 2024) | ✗ | ✗ | ✓ | ✗ | ✓ | ✓ | ✓ | 3.4k | 3.5k | 0.7k | 0.7k |
| Social-IQ (Zadeh et al., 2019) | ✓ | ✓ | ✓ | ✓ | ✓ | ✓ | ✗ | 1k | 6k | 0.3k | 1.5k |
| Social-IQ 2.0 (Wilf et al., 2023) | ✓ | ✓ | ✓ | ✓ | ✓ | ✓ | ✗ | 1.1k | 6.2k | 0.3k | 1.7k |
| *Our Work* | | | | | | | | | | | |
| **R³-Bench** | ✓ | ✓ | ✓ | ✓ | ✓ | ✓ | ✓ | - | - | 0.3k | 5.1k |
| **R³-FDT** | ✓ | ✓ | ✓ | ✓ | ✓ | ✓ | ✓ | 2.8k | 41k | - | - |

Table 1: Dataset comparison. "CC" means mental-physical causal chain. ✓ represents the dataset does not focus on a content. "# Video" and "# QA" represent the number of videos and question-answer pairs, respectively. The set sizes of Social-IQ are estimated using an 80%/20% ratio.

**F**oundation **D**ataset for **T**raining). They capture rich social interactions and include: (i) detailed social events, (ii) mental states and their transitions, and (iii) multi-step mental-physical causal chains. We evaluate state-of-the-art (SOTA) LVLMs on R³-Bench, and further fine-tune a 7B model on R³-FDT. The results demonstrate that: (i) current models still fall short of human-level social reasoning; (ii) our training data provides notable improvements across several benchmarks.

In summary, our contributions are three-fold: (i) we introduce R³-Bench, a novel benchmark with complete, fine-grained annotations for social reasoning; (ii) we show that SOTA LVLMs remain far from human-level performance on this benchmark; (iii) we construct R³-FDT using a scalable automated pipeline, offering valuable training data to improve LVLMs' social reasoning capabilities.

## 2 R³-BENCH: A HIGH-QUALITY TESTBED

We design R³-Bench in a natural, intuitive, and principled way, grounded in the foundational Theory of Mind (Premack & Woodruff, 1978), the belief-desire-intention (BDI) framework (Bratman, 1987), Bandura's social cognitive theory—particularly triadic reciprocal determinism (Bandura et al., 1986; Bandura, 1989)—and other modern studies in social cognition (Tomasello, 2010; Pearl, 2014; 2009; Reisenzein, 2006; 2009; Puica & Florea, 2013; Schlaffke et al., 2015; Fan et al., 2022). It systematically integrates both physical and mental dimensions of social interaction. We include key mental state variables (belief, intention, desire, and emotion) alongside observable physical variables (actions, expressions, dialogue, and other social cues). We further capture comprehensive and dynamic causal interactions among these variables, reflecting the intricate interplay between internal states and external behaviors and environments. This framework positions it as a valuable testbed for developing and evaluating computational models of social reasoning.

### 2.1 OVERALL DESIGN

R³-Bench is further divided into R³-Bench-Hard, which consists of sufficiently challenging questions originally designed by humans, and R³-Bench-DX, a diagnostic set derived from causal chains. It differs from traditional video question answering (VideoQA) benchmarks in the following aspects: (i) It includes fine-grained annotations of mental-physical causal chains; (ii) Question-answer (QA) pairs are derived based on these causal chains; (iii) It enables comprehensive evaluation of various social reasoning capabilities, including reasoning consistency via causal chains.

We illustrate the design and structure of R³-Bench-DX with an example in Figure 2a. Each video has one or more mental-physical causal chains. We denote a causal chain as $g \in \mathcal{G}$, and a subchain as $g^{\text{sub}} \in \mathcal{G}^{\text{sub}}$. Here, $\mathcal{G}$ and $\mathcal{G}^{\text{sub}}$ respectively represent all the causal chains and subchains. Note that a chain consists of multiple subchains. A subchain $g^{\text{sub}}$ comprises one result node $n^1$ and one or several reason nodes $\{n_i^0\}$. All reason nodes point to the result node via causal edges $\{e_i\}$. Every reason node $n_i^0$ is necessary to deduce the result node $n^1$.

For each causal chain $g$, we generate a set of related QA pairs according to the following rules:

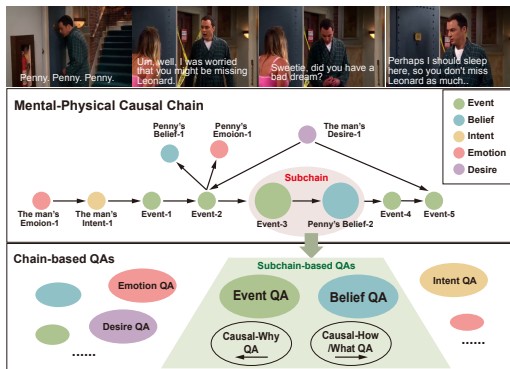

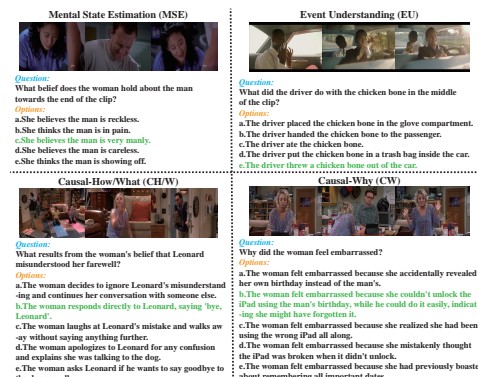

(a) General illustration of R³-Bench-DX design. As illustrated in the middle section, the man's mental states, Penny's mental states, and the events collectively form a mental-physical causal chain. The bottom section displays all QA pairs derived from it. The red circle highlights a subchain indicating that "Event-3 causes Penny to form Belief-2." The green trapezoid represents the corresponding derived QA pairs.

(b) Examples of each question type in R³-Bench-DX. The correct option is marked in green. In *MSE* example, the woman thinks that the man is manly while bandaging him. In *EU* example, the driver throws leftover chicken bones. In *CH/W* example, the woman realizes Leonard misunderstood her affectionate farewell, so she says goodbye to clarify. In *CW* example, the woman feels embarrassed by her forgetfulness.

Figure 2: Illustration of dataset design and question types of R³-Bench-DX.

(i) For each node $n \in g$, we generate an *Event Understanding* or *Mental State Estimation* QA pair, depending on its node content. The content of a node is either about an event or a mental state.

(ii) For each subchain $g^{\text{sub}}$, we generate a *Causal-Why* QA pair and a *Causal-How/What* QA pair, depending on whether the reasoning is abductive or deductive.

Therefore, there are four types of questions in R³-Bench-DX. We provide four examples selected from our dataset for the four question types respectively in Figure 2b:

***Event Understanding (EU).*** Understanding events is the premise of social reasoning. We generate a factual QA pair for each event node. For example, "What happens at the end of the clip?" and "What does Person B do when Person A reaches out her hands?".

***Mental State Estimation (MSE).*** Estimating mental states is a crucial aspect of social intelligence. We consider typical mental states including belief, intent, desire, and emotion. We generate an inferential QA pair for a mental state node, such as "How does someone feel at the end of the clip?"

***Causal-Why (CW) & Causal-How/What (CH/W).*** *CW* questions focus on abductive reasoning and inquire about causes of a given result—for example, the reasons why *the woman feels embarrassed* in the bottom-right case of Figure 2b. In contrast, *CH/W* questions emphasize deductive reasoning and ask about the effects of a cause, such as the outcome of *the woman's belief* in Figure 2b. To increase question diversity, we include variations such as "How" and "What" in phrasing. An example *Causal-How* QA pair is: *Question*: "How did Sheldon's revelation affect Penny's emotions?" *Answer*: "Sheldon's comment about the ring's true value led to Penny's disappointment."

## 2.2 DATASET CONSTRUCTION

R³-Bench is a comprehensive evaluation benchmark featuring high-quality and sufficiently challenging data. This is ensured through three aspects: (i) recruiting participants to independently submit diverse video sources—such as ads, short films, and real-life clips—that meet our criteria; (ii) using a powerful model to filter for difficult and unpolluted samples, meaning data unlikely to appear in standard pretraining sets; and (iii) involving domain experts to verify and annotate the content. To support this, we design a rigorous data construction pipeline illustrated in Figure 3.

**Human Data Collection.** To support high-quality human-designed QA pairs, we organize a challenge to collect data from crowdworkers and volunteers. To guide this process, we establish clear principles and standards: (i) Data composition: Each data sample includes a video clip, a question, four

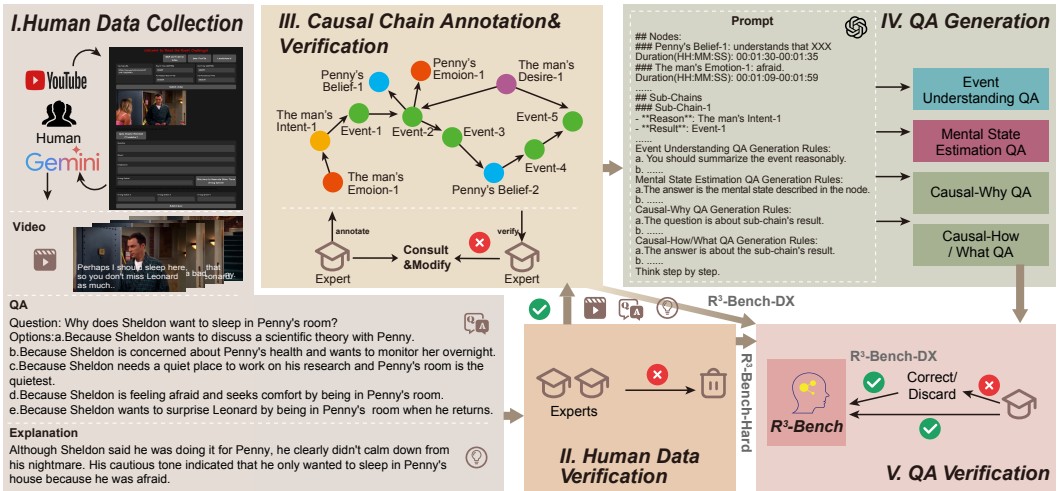

Figure 3: The R³-Bench construction pipeline is divided into five stages. i) **Human Data Collection.** Subjects select videos from YouTube and designed QA pairs with explanations. ii) **Human Data Verification.** Experts review the collected data and remove those that didn't meet standards. iii) **Causal Chain Annotation&Verification.** Experts annotate, verify, and refine the causal chains. iv) **QA Generation.** We utilize GPT-4o to generate QA pairs based on the causal chains. v) **QA Verification.** The experts from the previous stage verify and modify the generated QA pairs.

distractors, a correct answer, and a reasoning explanation. (ii) Strict standard: Questions must involve at least one mental state and emphasize causal relations. (iii) No specialized external knowledge required: Answers must be grounded primarily in observable cues and general common sense, without relying on specialized external knowledge. (iv) Explanation requirement: Each sample must include an explanation describing the reasoning process and underlying causal chain. (v) Data pollution prevention: We exclude any sample that Gemini 1.5 Pro (Team et al., 2024) answers correctly, ensuring the data remains unseen and challenging. This design ensures that the collected data reflects nuanced, causal, and human-level reasoning that current models are not yet capable of replicating.

**Human Data Verification.** While many submitted QA pairs successfully challenge Gemini 1.5 Pro, not all meet our quality standards. To ensure consistency, we conduct expert review involving annotators with backgrounds in cognitive science, linguistics, and AI. Each sample is assessed against our criteria—such as causal depth, mental state relevance, and clarity of explanation—by two independent reviewers. Only samples approved by both are retained in our benchmark, forming R³-Bench-Hard. Given the nuanced reasoning and social subtleties involved, such validation cannot be reliably automated, highlighting the necessity of human judgment for constructing a benchmark.

**Causal Chain Annotation & Verification.** To capture the reasoning behind each question, we conduct expert annotation of mental-physical causal chains. Annotators receive training based on strict principles and follow the structured format defined in Section 2.1. Each verified sample is annotated by one of its original reviewers. The annotation process consists of three steps: (i) reviewing the QA pair and its explanation, (ii) identifying key events and mental states as nodes, and (iii) linking these nodes into subchains according to causal relations. After annotation, the second expert conducts an independent review. If revisions are needed, both experts iterate to reach agreement on the final chain. This multi-stage, fine-grained process ensures that each sample reflects a coherent and interpretable causal reasoning flow grounded in rich multimodal context.

**QA Generation & Verification.** We use GPT-4o (Hurst et al., 2024) to generate QA pairs for each annotated causal chain, following the rules in Section 2.1. We invite the same experts from the causal chain annotation stage to verify the QA pairs generated from their own annotated chains. Each QA pair is checked against the following standards: (i) adherence to generation rules, (ii) accurate and unambiguous time references, (iii) full coverage of the corresponding node or subchain content, and (iv) only one correct answer among five options. If a QA pair fails to meet these criteria, experts revise it or discard it if correction is infeasible. This process yields 4,840 QA pairs, which, together with causal chains, constitute R³-Bench-DX. Detailed statistical analysis is provided in Section A.4.2.

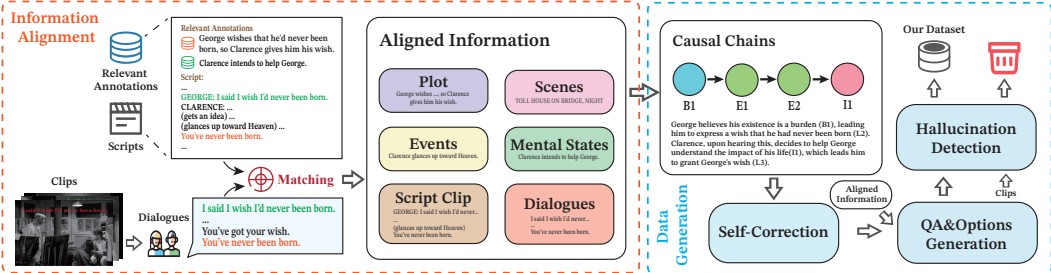

Figure 4: An automated pipeline for generating large-scale video social reasoning data using purely textual inputs. By aligning video clips with rich textual sources such as movie scripts and annotations, the system generates mental-physical causal chains and QA pairs via GPT-4o, and applies self-correction and hallucination detection for quality control.

# 3   R³-FDT: AUTOMATICALLY GENERATED DATASET WITH CAUSAL CHAINS

The development of VideoQA datasets for social reasoning is constrained by two significant challenges. First, there is a scarcity of high-quality videos that capture the complexity of social interactions. Second, the annotation process is prohibitively expensive, requiring intensive labor to infer nuanced mental states, followed by extensive verification to reduce ambiguity. These bottlenecks have impeded the development of foundational models in this domain. To overcome these limitations, we introduce an automated pipeline using human annotations and large models. As illustrated in Figure 4, it is designed to generate large-scale training datasets at a reduced cost while maintaining high data quality. Our methodology is centered on the following principles:

(i) Mitigation of Cross-Modal Reasoning Deficiencies: Current models often exhibit limitations such as inconsistencies and hallucinations when processing videos. Our pipeline circumvents these issues by utilizing textual descriptions of videos to enhance the credibility of generated content.

(ii) Assurance of High-Fidelity Annotations: The pipeline exclusively uses human-annotated descriptions that are aligned with videos. This ensures that the foundational data maintains a high degree of accuracy and reliability, consistent with expert annotation standards.

The proposed pipeline offers a cost-effective and scalable solution for generating high-quality training data, addressing a critical need for advancing foundation models in social reasoning.

## 3.1   INFORMATION ALIGNMENT

To enable large-scale generation of social reasoning data with mental-physical causal chains, we leverage movie data as a rich and structured source. Compared to open-domain videos, movies offer diverse content along with detailed scripts and annotations that describe both visual context and mental states. Therefore, they are naturally suitable for constructing training data that aligns structurally with our human-curated benchmark. We extract information from publicly available movie datasets such as MovieNet (Huang et al., 2020), MovieQA (Tapaswi et al., 2016), and CondensedMovies (Bain et al., 2020), as well as corresponding movie scripts. As shown in Figure 4, we extract key elements from each clip, including scene context, annotated events, mental states, and aligned dialogue. To ensure all generated QA pairs remain within the temporal boundaries of each clip, we use Whisper (Radford et al., 2023) to align scripts with detected dialogue. The alignment algorithm is detailed in Section A.5.1.

## 3.2   DATA GENERATION

**Causal Chains Generation.** Based on the aligned information, we prompt GPT-4o to infer causal links between events and mental states using contextual cues. In addition to leveraging existing annotations, the model can also identify latent mental states implied by character behavior and dialogue, thereby enriching the reasoning structure with nodes that are not explicitly labeled but contextually grounded. As illustrated in Figure 4, each causal chain is represented symbolically and paired with a natural language explanation. For each video clip, the model generates multiple chains

that reflect the most salient social interactions. This automated procedure retains the structural depth of human-curated reasoning while making it feasible to build diverse and scalable training data.

**Self-Correction.** To further improve data quality, we require GPT-4o to self-correct the causal chains generated in the previous stage. It needs to check the consistency between the symbolic and textual representations of chains, and remove redundant chains. In this stage, 6% of chains are removed. Although this accounts for only a small proportion, it is necessary to prevent the model from learning from erroneous noises.

**QA & Options Generation.** In accordance with the guidelines outlined in Section 2.1, a QA pair is generated for each node and subchain. To enhance the quality of the data, the inherent causal structure is utilized to create distractors that are both challenging and plausible. Specifically, employing the corrected causal chains and aligned information, GPT-4o generates four distractor options for each question, adhering to two primary principles: (i) the option is designed to be plausible from a common-sense perspective but lacks grounding in the video context; or (ii) the option is grounded in the video context but contradicts the established causal or mentalistic logic. These strategies necessitate a nuanced comprehension of both physical and social cues to arrive at the correct answer.

**Hallucination Detection.** We establish a hallucination detection stage to filter for high-quality QA pairs. Specifically, we input the original video and QA pairs into Gemini 2.5 Flash (Comanici et al., 2025), requiring it to perform a detailed hallucination analysis. During this process, the model is required to complete three key tasks: (i) determine if a QA pair aligns with the video, (ii) provide a detailed explanation, and (iii) output a confidence level. Only when a QA pair is determined to be completely free of hallucination do we retain it. Furthermore, we employ rule-based filtering to remove QA pairs with ambiguous speaker references. This strict quality control mechanism ensures the reliability of the final dataset, providing high-quality data for training a foundation model.

The resulting training set, $R^3$-FDT, consists of 41k QA pairs distributed across 2.8k videos. This constitutes a large-scale, causally structured dataset. Statistical analysis is provided in Section A.5.2.

## 4 EXPERIMENTS

Our experiments consist of two primary components. First, we evaluate a broad set of popular LVLMs on our benchmark $R^3$-Bench. Second, we train one of the best-performing models, Qwen2-VL-7B (Wang et al., 2024b) on our training set $R^3$-FDT. We then evaluate the trained model on $R^3$-Bench as well as on external social reasoning datasets to assess its generalization capabilities.

### 4.1 EVALUATION SETTINGS

We report the evaluation results under two experimental settings: a video-only ("-") setting and a video-plus-subtitle ("+Sub") setting. For models that cannot process video data directly, we uniformly sample 16 frames from each video to serve as input. For Gemini 1.5 which accept video inputs, we use raw videos and frames respectively. We found that Gemini 1.5 Pro's performance when processing raw video was inferior to its performance when processing individual frames, so we exclusively report on the performance of Gemini 2.5 Pro (Comanici et al., 2025) using frame-based inputs. We provide video subtitles generated using Whisper and incorporate them into the text prompts.

For $R^3$-Bench-DX, our evaluation approach consists of two dimensions: conventional accuracy metrics and our proposed consistency-based metrics. **(i) Accuracy Metrics.** We report the overall accuracy for each model, as well as accuracies across different question types. For the *MSE* category, we further break it down into four fine-grained subtypes: *Emotion*, *Belief*, *Intent*, and *Desire*. These distinctions help us identify which dimensions of social reasoning are more challenging. **(ii) Consistency Metrics.** Despite achieving relatively high overall accuracy, many models exhibit severe inconsistencies when answering logically related questions. For instance, a model might correctly answer the question "Why did A happen?" with "Because of B," yet fail when asked "Is B present in the video?" shortly after. Humans, in contrast, demonstrate coherent reasoning processes across multi-step chains, maintaining internal consistency throughout. To better capture this gap, we introduce two evaluation metrics: **Chain Consistency** and **Subchain Consistency**. They assess whether the model can consistently answer a set of questions derived from a single causal chain or its subcomponents.

A chain (or subchain) is marked "consistent" only if the model answers all associated questions correctly—partial correctness is not rewarded. Let $\mathcal{D}(g)$ denote the set of all data samples corresponding to a causal chain $g$, and $\mathcal{D}(g^{\text{sub}})$ for a subchain $g^{\text{sub}}$. Denote $|\mathcal{G}|$ as the number of causal chains, and $|\mathcal{G}^{\text{sub}}|$ for subchains. A data sample is denoted as $(v, q, a^{\text{gt}}, \mathcal{A})$, where $v$ is the video, $q$ is the question, $a^{\text{gt}}$ is the ground truth answer and $\mathcal{A}$ is the corresponding answer set. A model parameterized by $\theta$ selects an answer $a^*$ according to:

$$a^* = \arg\max_{a \in \mathcal{A}} P_\theta(a|v, q, \mathcal{A}) \tag{1}$$

We define **Chain Consistency** as:

$$\text{Cons}^{\text{c}} = \frac{\sum_{g \in \mathcal{G}} \prod_{(v,q,a^{\text{gt}},\mathcal{A}) \in \mathcal{D}(g)} \mathbb{I}(a^* = a^{\text{gt}})}{|\mathcal{G}|} \tag{2}$$

and **Subchain Consistency** as:

$$\text{Cons}^{\text{sc}} = \frac{\sum_{g^{\text{sub}} \in \mathcal{G}^{\text{sub}}} \prod_{(v,q,a^{\text{gt}},\mathcal{A}) \in \mathcal{D}(g^{\text{sub}})} \mathbb{I}(a^* = a^{\text{gt}})}{|\mathcal{G}^{\text{sub}}|} \tag{3}$$

These metrics go beyond surface-level accuracy by evaluating whether a model can sustain coherent reasoning across entire social interaction chains. They are particularly effective at revealing hidden weaknesses in models, where high average accuracy may mask fragmented or inconsistent behavior.

## 4.2 EVALUATION RESULTS AND KEY INSIGHTS

As shown in Table 2, current models still lag significantly behind humans on challenging social reasoning tasks. Gemini 2.5 Pro achieves the best performance among the models with 59.18%, yet remains far below the human level of 80.06%. Many models have accuracies below or around 30%, indicating that most models perform close to random guessing on R$^3$-Bench-Hard and fail to demonstrate effective, high-level social reasoning capabilities.

As illustrated in Table 3, the evaluation results on R$^3$-Bench-DX yield two primary insights. First, models find reasoning about mental states substantially more challenging than understanding factual events. For example, in the "+Sub" setting, GPT-4o's accuracy drops from 89.77% on *EU* to 72.44% on *MSE*, highlighting the difficulty of inferring abstract internal states. Second, a critical limitation is the models' failure to maintain coherent reasoning, despite high accuracy on individual questions. This disconnect is evident across all models. For instance, in the "+Sub" setting, GPT-4o scores 82.64% in overall accuracy but only 25.36% in chain consistency ($\text{Cons}^{\text{c}}$). Gemini 2.5 Pro shows a similar gap, with 86.34% accuracy versus 36.60% consistency. This "high accuracy, low consistency" paradox reveals that models lack a holistic, structured understanding of social interactions, a weakness that single-question accuracy metrics fail to capture.

Table 2: R$^3$-Bench-Hard evaluation results (%). All models are evaluated under video-plus-subtitle setting. "+ Ours-FT (SFT)" and "+ Ours-FT (RLFT)" denote SFT and GRPO training on R$^3$-FDT respectively.

| Model | Overall |
|---|---|
| Random | 20 |
| Idefics2-8B (Laurençon et al., 2024b) | 15.19 |
| PLLaVA-7B (Xu et al., 2024) | 17.09 |
| Video-LLaVA (Lin et al., 2023) | 18.35 |
| PLLaVA-13B (Xu et al., 2024) | 20.25 |
| Phi-3.5-Vision (Abdin et al., 2024) | 23.73 |
| Idefics3-8B-Llama3 (Laurençon et al., 2024a) | 24.37 |
| InternVL2-8B (OpenGVLab, 2024) | 24.68 |
| InternVL2-26B (OpenGVLab, 2024) | 24.68 |
| Gemini 1.5 Flash (Team et al., 2024) (Video) | 28.48 |
| mPLUG-Owl3 (Ye et al., 2024) | 29.11 |
| PLLaVA-34B (Xu et al., 2024) | 30.06 |
| Gemini 1.5 Flash (Frames) (Team et al., 2024) | 30.38 |
| GPT-4o Mini (Hurst et al., 2024) | 30.70 |
| InternVL2-76B (OpenGVLab, 2024) | 31.96 |
| Gemini 1.5 Pro (Team et al., 2024) (Video) | 34.81 |
| Gemini 1.5 Pro (Team et al., 2024) (Frames) | 39.24 |
| GPT-4o (Hurst et al., 2024) | 48.73 |
| Gemini 2.5 Pro (Comanici et al., 2025) (Frames) | **59.18** |
| Qwen2-VL-7B (Wang et al., 2024b) | 31.96 |
| Qwen2-VL-7B + Ours-FT (RLFT) | 39.87 |
| Qwen2-VL-7B + Ours-FT (SFT) | 42.09 |
| Human | **80.06** |

## 4.3 FINE-GRAINED ANALYSIS ON COGNITIVE DIMENSIONS

To delve deeper into the specific failures of current models, we move beyond a monolithic performance metric on R$^3$-Bench-Hard to a fine-grained analysis based on six cognitive dimensions. Figure 5a presents the performance breakdown of various models across these dimensions, benchmarked against human performance. The results reveal a clear and consistent pattern of deficiencies.

Table 3: $R^3$-Bench-DX evaluation results (%). *MSE*: *Mental State Estimation*. *EU*: *Event Understanding*. *CW*: *Causal-Why*. *CH/W*: *Causal-How/What*. $\text{Cons}^c$: Chain Consistency. $\text{Cons}^{sc}$: Subchain Consistency. "-" denotes the video-only setting, and "+Sub" denotes the video-plus-subtitle setting. "+ Ours-FT (SFT)" and "+ Ours-FT (RLFT)" denote SFT and GRPO training on $R^3$-FDT respectively.

| Model | Setting | MSE | | | | | EU | CW | CH/W | Overall | $\text{Cons}^c$ | $\text{Cons}^{sc}$ |
|---|---|---|---|---|---|---|---|---|---|---|---|---|
| | | *Emotion* | *Belief* | *Intent* | *Desire* | *Overall* | | | | | | |
| Random | - | 20 | 20 | 20 | 20 | 20 | 20 | 20 | 20 | 20 | - | - |
| Video-LLaVA | - | 18.03 | 18.69 | 20.22 | 16.67 | 18.82 | 20.26 | 20.57 | 21.58 | 20.33 | *0.00* | *0.14* |
| (Lin et al., 2023) | +Sub | 19.08 | 19.63 | 24.38 | 21.43 | 20.90 | 28.28 | 21.64 | 22.88 | 23.14 | *0.00* | *0.36* |
| PLLaVA-7B | - | 28.93 | 26.48 | 27.98 | 14.29 | 27.48 | 29.29 | 31.10 | 32.74 | 30.25 | *0.29* | *1.42* |
| (Xu et al., 2024) | + Sub | 22.64 | 22.43 | 26.59 | 23.81 | 23.81 | 36.21 | 29.25 | 29.02 | 29.28 | *0.00* | *1.21* |
| Idefics2-8B | - | 10.48 | 8.41 | 10.80 | 2.38 | 9.74 | 8.93 | 8.90 | 11.48 | 9.77 | *0.00* | *0.07* |
| (Laurençon et al., 2024b) | + Sub | 21.80 | 21.50 | 22.99 | 19.05 | 21.98 | 20.56 | 22.99 | 22.64 | 22.15 | *0.29* | *1.14* |
| PLLaVA-13B | - | 23.48 | 22.43 | 22.71 | 16.67 | 22.73 | 23.77 | 33.10 | 30.72 | 28.00 | *0.00* | *0.64* |
| (Xu et al., 2024) | + Sub | 27.04 | 30.84 | 26.04 | 26.19 | 27.73 | 37.61 | 38.58 | 35.57 | 34.92 | *0.58* | *2.56* |
| GPT-4o Mini | - | 43.19 | 57.63 | 55.40 | 52.38 | 51.04 | 62.39 | 73.52 | 69.36 | 64.59 | *6.05* | *18.92* |
| (Hurst et al., 2024) | + Sub | 43.19 | 57.63 | 55.40 | 52.38 | 51.04 | 62.39 | 73.52 | 69.52 | 64.63 | *6.05* | *18.99* |
| InternVL2-8B | - | 47.17 | 51.09 | 42.11 | 47.62 | 46.71 | 46.94 | 61.78 | 58.69 | 54.19 | *3.17* | *11.17* |
| (OpenGVLab, 2024) | + Sub | 49.90 | 68.54 | 54.57 | 52.38 | 56.37 | 73.22 | 73.38 | 68.31 | 67.83 | *8.06* | *25.04* |
| Idefics3-8B-Llama3 | - | 40.88 | 57.94 | 48.75 | 35.71 | 47.63 | 51.15 | 63.70 | 62.49 | 56.82 | *3.17* | *12.02* |
| (Laurençon et al., 2024a) | + Sub | 45.49 | 71.65 | 55.12 | 54.76 | 55.70 | 74.32 | 74.45 | 69.44 | 68.49 | *8.07* | *23.83* |
| Phi-3.5-Vision | - | 48.85 | 59.50 | 49.86 | 35.71 | 51.54 | 53.46 | 72.31 | 70.74 | 62.87 | *4.90* | *16.36* |
| (Abdin et al., 2024) | + Sub | 46.75 | 62.62 | 53.46 | 40.48 | 52.79 | 68.71 | 76.16 | 72.92 | 68.00 | *8.65* | *23.47* |
| mPLUG-Owl3 | - | 48.01 | 59.50 | 42.94 | 45.24 | 49.46 | 52.46 | 67.90 | 63.22 | 58.95 | *2.88* | *13.66* |
| (Ye et al., 2024) | + Sub | 51.15 | 71.03 | 50.97 | 50.00 | 56.37 | 75.03 | 74.66 | 70.82 | 69.21 | *9.22* | *25.39* |
| Gemini 1.5 Flash(Frames) | - | 48.43 | 65.73 | 60.94 | 52.38 | 56.95 | 67.60 | 73.10 | 70.57 | 67.31 | *6.05* | *23.33* |
| (Team et al., 2024) | + Sub | 49.90 | 72.90 | 65.37 | 59.52 | 61.03 | 80.84 | 76.87 | 74.78 | 73.22 | *11.24* | *33.14* |
| Gemini 1.5 Flash (Video) | - | 37.95 | 60.44 | 53.74 | 57.14 | 49.38 | 69.21 | 70.39 | 65.48 | 63.68 | *8.65* | *22.62* |
| (Team et al., 2024) | + Sub | 47.38 | 72.59 | 64.27 | 64.29 | 59.78 | 80.34 | 78.58 | 72.91 | 72.91 | *11.82* | *30.73* |
| InternVL2-26B | - | 42.14 | 49.22 | 45.43 | 45.24 | 45.13 | 47.44 | 59.86 | 57.56 | 53.06 | *3.17* | *10.60* |
| (OpenGVLab, 2024) | + Sub | 46.12 | 71.96 | 58.73 | 57.14 | 57.20 | 73.62 | 74.45 | 71.22 | 69.17 | *13.26* | *27.03* |
| PLLaVA-34B | - | 49.90 | 59.81 | 55.40 | 54.76 | 54.37 | 52.46 | 69.54 | 70.57 | 62.52 | *5.19* | *16.22* |
| (Xu et al., 2024) | + Sub | 53.67 | 70.40 | 68.14 | 69.05 | 63.03 | 78.44 | 77.30 | 78.42 | 74.28 | *14.12* | *33.50* |
| InternVL2-76B | - | 42.35 | 63.86 | 48.48 | 45.24 | 50.04 | 57.87 | 69.11 | 66.61 | 61.43 | *5.75* | *15.58* |
| (OpenGVLab, 2024) | + Sub | 53.46 | 75.70 | 65.10 | 64.29 | 63.28 | 81.34 | 79.93 | 76.39 | 75.19 | *17.29* | *35.99* |
| Gemini 1.5 Pro (Video) | - | 50.31 | 75.08 | 74.79 | 69.05 | 64.95 | 84.65 | 80.13 | 80.52 | 77.39 | *15.85* | *38.69* |
| (Team et al., 2024) | + Sub | 56.39 | 74.77 | 74.79 | 73.81 | 67.44 | 84.55 | 80.50 | 77.45 | 77.31 | *19.60* | *41.61* |
| Gemini 1.5 Pro (Frames) | - | 44.44 | 65.11 | 67.31 | 54.76 | 57.20 | 68.61 | 76.30 | 73.57 | 69.28 | *8.93* | *23.97* |
| (Team et al., 2024) | + Sub | 53.67 | 74.45 | 77.56 | 71.43 | 67.03 | 85.36 | 81.71 | 78.66 | 78.04 | *20.75* | *44.10* |
| GPT-4o | - | 60.17 | 80.37 | 79.50 | 76.19 | 71.94 | 89.17 | 85.84 | 82.54 | 82.23 | *25.07* | *47.94* |
| (Hurst et al., 2024) | + Sub | 61.01 | 80.69 | 79.78 | 76.19 | 72.44 | 89.77 | 86.05 | 82.94 | 82.64 | *25.36* | *48.93* |
| Gemini 2.5 Pro (Frames) | - | 56.81 | 83.49 | 80.06 | 73.81 | 77.12 | 83.85 | 85.84 | 81.57 | 80.79 | *24.50* | *43.53* |
| (Comanici et al., 2025) | + Sub | 65.83 | **88.47** | **85.60** | **88.10** | **85.17** | 93.08 | 88.33 | 86.18 | 86.34 | ***36.60*** | *58.82* |
| Qwen2-VL-7B | - | 58.07 | 60.75 | 55.40 | 54.76 | 58.51 | 59.28 | 72.31 | 71.87 | 65.93 | *5.48* | *19.63* |
| (Wang et al., 2024b) | + Sub | 58.91 | 70.40 | 63.43 | 59.52 | 70.02 | 78.03 | 79.22 | 78.66 | 74.90 | *12.68* | *33.85* |
| | + Ours-FT (RLFT) | 77.36 | 83.80 | 80.33 | 76.19 | 83.67 | 88.16 | 91.25 | 87.63 | 86.88 | *29.11* | *55.83* |
| | + Ours-FT (SFT) | **80.08** | 87.85 | 83.66 | **88.10** | 83.51 | 89.57 | **92.81** | **88.52** | **88.74** | *35.73* | ***60.95*** |
| Human | | **93.55** | **92.50** | **92.31** | **100.00** | **92.98** | **93.91** | **90.21** | **92.37** | **92.24** | **60.47** | **75.52** |

Our analysis yields several key insights. First, the most profound weaknesses are exposed in tasks requiring detection of **Contradiction in Words vs Behavior** and pragmatic inference for **Implication / Reading Between Lines**. Even the best-performing model, Gemini 2.5 Pro, lags significantly behind human accuracy (e.g., 48.2% vs. 78.8% in contradiction detection). This reveals a fundamental deficit in handling modality conflicts and performing pragmatic inference, suggesting models favor literal interpretations over grasping nuanced social contexts involving sarcasm or deceit.

Second, the results point to architectural limitations. On one hand, models lack systematic modeling of temporal event structures and evolving mental states. This explains why even Gemini 2.5 Pro and GPT-4o fall short of human robustness (e.g., 77.4% vs. 87.1%) on **Plot Twist → Mental State Change** tasks. On the other hand, predominant architectures often align multimodal features only superficially. Consequently, when tasks demand deep integration of distinct modalities to uncover underlying meaning, particularly within the **Must Combine Visual and Verbal Cues** dimension, their performance drops sharply (to 58.6% and 50.5%, respectively).

Finally, the **Imagination Beyond Video** category serves as a powerful diagnostic. Here, the performance of top-tier models like Gemini 2.5 Pro and GPT-4o (both at 68.8%) closely approaches the human baseline (75.0%). This highlights the inherent advantage of a powerful linguistic world model

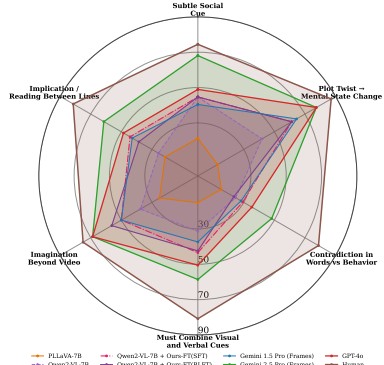
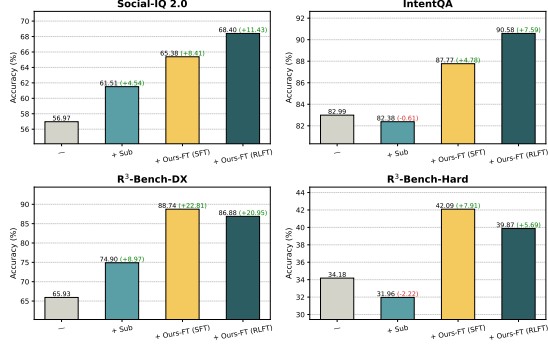

(a) Radar chart of model accuracy (%) across six cognitive dimensions. Results highlight significant gap between models and human-level reasoning, particularly in tasks requiring detection of multi-modal contradictions and pragmatic inference.

(b) **Accuracy (%)** on four video social reasoning benchmarks before and after fine-tuning Qwen2-VL-7B. "-" = video-only setting, "+ Sub" = video-plus-subtitle setting, "+ Ours-FT (SFT)" = SFT on $R^3$-FDT, and "+ Ours-FT (RLFT)" = GRPO training on $R^3$-FDT.

Figure 5: Comparison of cognitive and social reasoning capabilities. (a) Model accuracy across six cognitive dimensions on $R^3$-Bench-Hard. (b) Improvements on relevant benchmarks after fine-tuning.

for tasks that require reasoning beyond direct perceptual evidence, affirming that strong language priors are crucial for imaginative inference.

In summary, this analysis indicates that future advancements in multimodal social reasoning must prioritize: (i) robustly modeling mental states and motivations; (ii) enhancing pragmatic inference for non-literal language understanding; and (iii) improving resolution of contradictions across modalities.

## 4.4 TRAINING ON $R^3$-FDT

To address previously identified challenges in consistency and multi-step reasoning, we apply reinforcement learning fine-tuning (RLFT) using the GRPO algorithm (Shao et al., 2024) and supervised fine-tuning (SFT) on a sampled subset of 13k QA pairs from $R^3$-FDT, with subtitles incorporated into prompts. For RLFT, the reward signal is formulated as a multiple-choice task. For SFT, we convert 10% of the training samples into an open-ended format.

Although the training and test sets differ substantially in video domain—our training set consists of curated movie clips, while the test benchmark includes diverse YouTube-style content—they share a unified structure grounded in causal chains. This structural alignment allows the model to benefit directly from training signals, leading to performance improvements closely aligned with evaluation objectives. As shown in Figure 5b, training on our training subset yields substantial gains. Specifically, RLFT achieves improvements of +11.98% on $R^3$-Bench-DX and +5.69% on $R^3$-Bench-Hard, while SFT shows increases of +13.84% and +7.91% respectively. Further, as reported in Tables 2 and 3, the fine-tuned models outperform Gemini 2.5 Pro on $R^3$-Bench-DX and surpasses Gemini 1.5 Pro on $R^3$-Bench-Hard, demonstrating enhanced social reasoning across both question styles.

To assess generalization, we also evaluate the model on two external video social reasoning datasets: Social-IQ 2.0 and IntentQA. As shown in Figure 5b, RLFT yields improvements of 6.89% and 7.59%, while SFT results in gains of 3.87% and 4.78%, respectively, confirming the transferability of reasoning capabilities acquired from our training subset.

## 5 CONCLUSION

In this work, we model the causal flow between mental states and physical events in complex social interactions through mental-physical causal chains. Based on this structure, we construct $R^3$-Bench for evaluation and $R^3$-FDT for model development. $R^3$-Bench highlights that current SOTA LVLMs still exhibit deficiencies in social reasoning capabilities. Fine-tuning Qwen2-VL-7B on $R^3$-FDT leads to notable gains across multiple relevant benchmarks, demonstrating its value. We hope they can contribute to the advancement of multimodal social intelligence and encourage further exploration.

## ETHICS STATEMENT

Our study involves human participants. Informed consent was obtained from all participants prior to data collection. The released datasets have been carefully anonymized to remove any personally identifiable information, and participants were informed that their data may be shared for research purposes. To mitigate potential misuse, the datasets are distributed under a research-only license, and documentation describing appropriate usage scenarios is provided. We believe that the potential benefits of these datasets for advancing research outweigh possible risks, and we have taken steps to minimize privacy, security, and fairness concerns in accordance with the ICLR Code of Ethics.

## REPRODUCIBILITY STATEMENT

We have taken multiple measures to ensure the reproducibility of our results. We provide a detailed description of our data collection method in Section 2 and Section 3, and report implementation details in Section A. To further support reproducibility, we release our datasets, code, and models.

## ACKNOWLEDGEMENT

This work is supported by National Natural Science Foundation of China (62406031), and National Science and Technology Major Project (2022ZD0114900). Yisen Wang is supported by National Natural Science Foundation of China (62376010, 92370129), Beijing Major Science and Technology Project under Contract No. Z251100008425006, Beijing Nova Program (20230484344, 20240484642), and State Key Laboratory of General Artificial Intelligence. Thanks to Ms. Zhen Chen for creating the beautiful figures.

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

# A APPENDIX

## A.1 THE USE OF LARGE LANGUAGE MODELS (LLMS)

We use LLMs in the following aspects: (i) We utilized LLMs in the dataset construction process; (ii) We evaluated task performance on several LLMs; (iii) We fine-tuned a LLM; (iv) We used LLMs to assist us with paper writing slightly.

## A.2 RELATED WORK

### A.2.1 SOCIAL REASONING IN LARGE LANGUAGE MODELS

Recent studies have explored the extent to which large language models (LLMs) can perform social reasoning, such as attributing beliefs (Lu et al., 2025), recognizing emotions, or inferring intentions (Shapira et al., 2023; Kim et al., 2023). Hi-ToM (He et al., 2023), SimpleToM (Gu et al., 2024), and BigToM (Gandhi et al., 2023) attempt to evaluate Theory-of-Mind (ToM) reasoning in LLMs. They establish crucial frameworks and standards for evaluating high-order ToM reasoning capabilities of LLMs, thereby providing valuable insights and directions for improvement. Nevertheless, their scope is limited, as they concentrate exclusively on two kinds of mental states (beliefs and intentions) and are restricted to textual contexts, neglecting the assessment of multimodal social cues and reasoning. Benchmarks like SocialIQa (Sap et al., 2019) and Social Chemistry 101 (Hwang et al., 2021) provide scenarios where models must understand social norms or moral judgments. However, these benchmarks rely solely on static text, lacking the multimodal and dynamic context typical of real-life social interactions. These limitations motivate the need for evaluation environments grounded in realistic, multimodal social interactions. Our datasets address this gap by offering richly annotated video scenarios designed to test multi-step reasoning over belief, desire, intent, and emotion in context.

### A.2.2 MULTIMODAL SOCIAL INTELLIGENCE AND VISION-LANGUAGE MODELS

Large vision-language models (LVLMs) such as Flamingo (Alayrac et al., 2022), VILA (Chen et al., 2023), and Video-LLaVA (Lin et al., 2023) have shown strong performance in general-purpose video-language tasks. However, their ability to perform fine-grained social reasoning (Fan et al., 2018; 2019) remains limited. Studies have found that these models often over-rely on language priors and fail to leverage visual evidence when answering socially grounded questions (Chen et al., 2024). Datasets like Social-IQ 2.0 (Wilf et al., 2023), VLEP (Lei et al., 2020), and SODA (Wang et al., 2023) introduce social content into video question answering (VideoQA), but few offer systematic annotations of mental-state transitions or causal relations. Additionally, models rarely capture nuanced cues like interpersonal gaze, tone, or posture, which are essential for deeper social inference (Wei et al., 2024). Our datasets complement these efforts by providing a scalable training set and a fine-grained benchmark with explicit labels for mental-state categories and their causal links in socially rich scenarios.

### A.2.3 THEORY OF MIND IN VIDEO UNDERSTANDING

ToM — the ability to attribute mental states to others — has been a long-standing challenge in AI. Recent work like Watch-and-Help (Puig et al., 2020), Smart Help (Cao et al., 2024), NoPa (Puig et al., 2023), and Generative Agents (Park et al., 2023) explores ToM in simulation-based or scripted agent settings. MMToM-QA (Jin et al., 2024) represents a step toward real-world ToM inference, offering multimodal video-based questions about beliefs and desires. Fan et al. (2021) model belief dynamics directly from nonverbal video signals. Yet, most of these benchmarks focus on short clips, isolated mental states, or handcrafted settings. IntentQA (Li et al., 2023) explores intention inference in VideoQA, but it does not capture full mental-state causal chains. Our proposed benchmark $R^3$-Bench explicitly models belief, intent, desire, and emotion along multi-step causal paths, enabling a more complete and diagnostic evaluation of ToM-like reasoning in LVLMs under naturalistic, temporally extended scenarios.

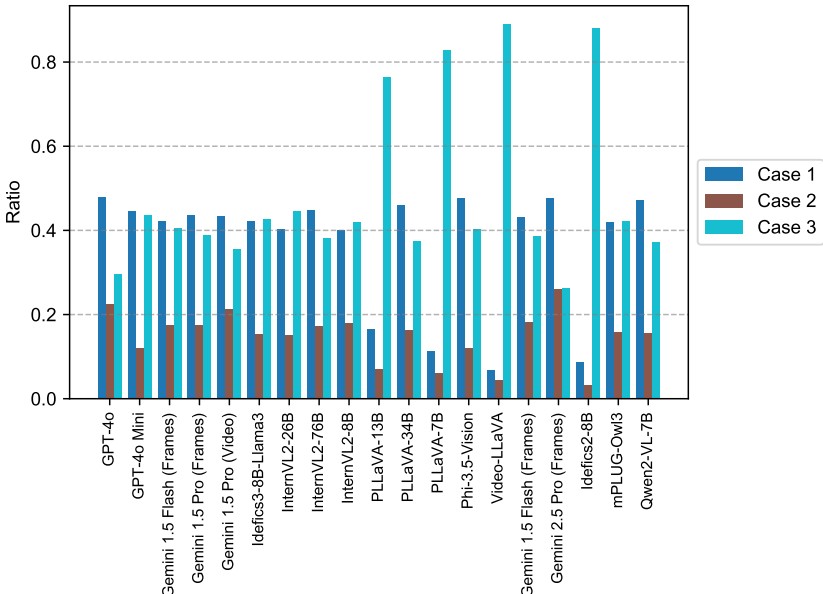

Figure 6: The ratio of the three cases. Case 1: The model answers a high-level causal question correctly without correctly answering the preceding, simpler event/mental state questions in the sub-chain. Case 2: The model answers basic event/mental state questions correctly but fails at the more complex, high-level causal questions. Case 3: The model fails to correctly answer both the basic event/mental state questions and the high-level causal questions.

## A.3 ADDITIONAL EXPERIMENTS AND ANALYSIS

### A.3.1 ANALYSIS OF THE CAUSES OF LOW CONSISTENCY

Inconsistent sub-chains can be categorized into the following scenarios:

(i) **Case 1**: The model answers high-level causal questions correctly without correctly answering the preceding, simpler event/mental state questions in the subchain;

(ii) **Case 2**: The model answers basic event/mental state questions correctly but fails at the more complex, high-level causal questions;

(iii) **Case 3**: The model fails to correctly answer both the basic event/mental state questions and the high-level causal questions.

Regarding Case 1, high-level causal questions are inherently hackable (note that this is an inherent challenge in evaluating high-level reasoning, rather than a specific defect in our dataset's design). Models can employ heuristics or shortcuts—such as relying on common sense or eliminating options that did not occur in the video—to answer these questions correctly. This "hijacking" phenomenon typically goes undetected in other datasets. However, by leveraging causal chains and consistency metrics, we can easily identify it, thereby providing a more reliable evaluation.

We analyze the proportions of these three cases to determine the primary causes of low consistency. As shown in Figure 6, for lower-performance models (e.g., PLLaVA, Video-LLaVA), Case 3 accounts for the largest proportion, indicating severe deficiencies in both fundamental capabilities and high-level reasoning. Conversely, for high-performance models (e.g., Gemini, GPT), the combined proportion of Case 1 and Case 2 increases significantly, with Case 1 being the most dominant. This highlights the prevalence and severity of the aforementioned "hijacking" phenomenon. While other datasets might overlook this issue and report overestimated causal reasoning capabilities, our consistency metric accurately detects it—allowing us to distinguish between flawed/hackable questions (Case 1) and genuine model reasoning deficits (Case 2)—and offers a more reliable assessment, fully demonstrating the significant value of $R^3$-Bench.

### A.3.2 DETAILED ANALYSIS USING FURTHER CLASSIFICATION OF SUBCHAINS

**Further Classification of Subchains**    We further divide subchains (stand for social causality) into six categories. Category I is MS → E, which means that an event is the result of one or more mental states. Category II is MS & E → E, which means that an event is the result of the combination of one or more mental states and one or more events. Category III is E → E, which means that an event is the result of one or more events. Category IV is E → MS, which means that a mental state is the result of one or more events. Category V is E & MS → MS, which means that a mental state is the result of the combination of one or more events and one or more mental states. Category VI is MS → MS, which means that a mental state is the result of one or more mental states. Through division, we can delve deeper into discussing large vision-language models' (LVLMs') performance on various types of social causality and analyze their strengths and weaknesses.

We quantify the number of *CW* QA pairs and *CH/W* QA pairs according to each of the six categories of subchains, as presented in Table 4. Among these, the questions that involve mutual reasoning between MS and E (i.e., Categories I and IV) are the most prevalent. In contrast, questions that employ both E and MS to reason about MS (i.e., Category V) are the least numerous and constitute the most challenging subset of QA pairs.

Table 4: Total quantities per category. The table categorizes social causality into six categories: I. MS → E. II. MS & E → E. III. E → E. IV. E → MS. V. E & MS → MS. VI. MS → MS.

| Category | Total Subchains | *CW* | *CH/W* | Total |
|---|---|---|---|---|
| I | 434 | 434 | 392 | 826 |
| II | 199 | 199 | 170 | 369 |
| III | 96 | 96 | 91 | 187 |
| IV | 464 | 463 | 403 | 863 |
| V | 79 | 79 | 65 | 144 |
| VI | 134 | 134 | 116 | 250 |
| **Total** | **1406** | **1405** | **1237** | **2642** |

**Results and Analysis**    We conduct a further analysis of the R$^3$-Bench-DX by categorizing them according to the six subchain classifications outlined in Section A.3.2. As shown in Table 5, we use the same models and settings reported in the main paper to evaluate our dataset. These settings included scenarios without subtitles ("-") and with subtitles ("+ Sub"). The metrics we report are similar to those in the main paper, encompassing both the accuracy of the QA pairs and the subchain consistency (Cons$^{sc}$) for each subchain. However, we further disaggregated the performance statistics based on the six subchain categories. This detailed analysis revealed several intriguing insights:

**Subchain Consistency for Mental State Reasoning:** Reasoning about events generally exhibits higher subchain consistency compared to reasoning about mental states. Specifically, Categories I, II, and III, which involve inferring events from events or mental states, demonstrate higher performance than Categories IV, V, and VI, which involve inferring mental states from events or mental states. Notably, event-to-event reasoning (Category III) achieves the highest accuracy and subchain consistency, significantly outperforming the other categories that involve mental state reasoning (Categories I, II, IV, V, and VI). This indicates that current LVLMs are more adept at factual causal reasoning than at inferring mental states, which remains more challenging.

**Difficulty in Cross-Domain Reasoning:** Inferring mental states from events or mental states is typically more challenging than inferring events from events or mental states. In Categories I and II, the core of the reasoning process is based on mental states (MS), with events (E) serving as auxiliary information to aid in accurately inferring the events. This auxiliary role of events facilitates correct reasoning, resulting in Categories I (MS inferring E) and II (MS & E inferring E) not exhibiting a pronounced increase in difficulty. Conversely, Category V, where both events (E) and mental states (MS) are used to infer mental states (MS), is the most difficult, significantly more so than Category IV, which involves events (E) inferring mental states (MS). In these categories, the mental states are the primary focus of inference, and the inclusion of additional mental state information introduces

Table 5: Additional results on R$^3$-Bench-DX. The table categorizes social causality into six categories: I. MS → E, II. MS & E → E, III. E → E, IV. E → MS, V. E & MS → MS, VI. MS → MS. **All values are reported as percentages (without % symbols), and the** Cons$^{sc}$ **columns are shown in bold.** "+ Ours-FT (SFT)" denotes supervised fine-tuning on R$^3$-FDT, while "+ Ours-FT (RLFT)" denotes GRPO training on R$^3$-FDT.

| Setting | Method | I CW | CH/W | Cons$^{sc}$ | II CW | CH/W | Cons$^{sc}$ | III CW | CH/W | Cons$^{sc}$ | IV CW | CH/W | Cons$^{sc}$ | V CW | CH/W | Cons$^{sc}$ | VI CW | CH/W | Cons$^{sc}$ |
|---|---|---|---|---|---|---|---|---|---|---|---|---|---|---|---|---|---|---|---|
| Video-LLaVA | - | 20.51 | 19.90 | **0.00** | 22.61 | 22.35 | **0.00** | 12.50 | 9.89 | **0.00** | 21.81 | 23.33 | **0.22** | 22.78 | 29.23 | **0.00** | 17.16 | 25.00 | **0.75** |
| | + Sub | 21.20 | 22.96 | **0.23** | 24.12 | 23.53 | **0.00** | 13.54 | 12.09 | **0.00** | 23.76 | 23.82 | **0.86** | 22.78 | 27.69 | **0.00** | 16.42 | 24.14 | **0.00** |
| Idefics2-8B | - | 9.68 | 10.20 | **0.00** | 10.05 | 14.12 | **0.00** | 11.46 | 12.09 | **0.00** | 7.56 | 11.66 | **0.00** | 8.86 | 10.77 | **0.00** | 7.46 | 11.21 | **0.75** |
| | + Sub | 23.96 | 23.98 | **1.15** | 22.11 | 22.94 | **1.01** | 28.12 | 21.98 | **1.04** | 23.33 | 21.34 | **1.72** | 20.25 | 20.00 | **0.00** | 17.91 | 24.14 | **0.00** |
| mPLUG-Owl3 | - | 59.91 | 66.07 | **13.36** | 71.36 | 65.29 | **9.55** | 77.08 | 65.93 | **18.75** | 73.00 | 60.05 | **17.03** | 60.76 | 66.15 | **3.80** | 68.66 | 57.76 | **11.19** |
| | + Sub | 65.67 | 75.77 | **23.04** | 76.88 | 76.47 | **24.12** | 87.50 | 80.22 | **45.83** | 79.05 | 63.03 | **28.66** | 77.22 | 70.77 | **13.92** | 74.63 | 65.52 | **15.67** |
| Phi-3.5-Vision | - | 65.67 | 74.23 | **14.06** | 78.89 | 74.12 | **11.56** | 78.12 | 64.84 | **15.62** | 74.51 | 66.75 | **20.91** | 78.48 | 78.33 | **6.33** | 68.66 | 69.83 | **21.64** |
| | + Sub | 68.66 | 77.55 | **20.05** | 82.91 | 82.94 | **19.60** | 80.21 | 64.84 | **27.08** | 79.05 | 67.74 | **28.66** | 78.48 | 69.23 | **15.19** | 76.12 | 68.97 | **24.63** |
| Idefics3-8B-Llama3 | - | 57.14 | 64.03 | **11.29** | 68.34 | 64.71 | **9.05** | 77.08 | 57.14 | **12.50** | 65.23 | 60.79 | **14.01** | 60.76 | 67.69 | **5.06** | 64.18 | 61.21 | **15.67** |
| | + Sub | 67.74 | 75.51 | **22.35** | 77.89 | 79.41 | **23.62** | 83.33 | 68.13 | **35.42** | 76.03 | 62.53 | **24.78** | 81.01 | 64.62 | **12.66** | 75.37 | 62.07 | **23.88** |
| PLLaVA-7B | - | 31.34 | 33.16 | **1.15** | 33.67 | 32.94 | **0.50** | 30.21 | 23.08 | **1.04** | 31.10 | 34.00 | **2.37** | 30.38 | 40.00 | **0.00** | 26.87 | 30.17 | **1.49** |
| | + Sub | 29.03 | 30.10 | **0.69** | 28.64 | 28.82 | **0.00** | 25.00 | 20.88 | **2.08** | 31.10 | 30.02 | **2.16** | 31.65 | 29.23 | **1.27** | 25.37 | 28.45 | **0.75** |
| PLLaVA-13B | - | 31.80 | 27.30 | **0.23** | 33.17 | 32.94 | **0.00** | 28.12 | 21.98 | **1.04** | 34.77 | 33.75 | **1.08** | 37.97 | 36.92 | **0.00** | 31.34 | 31.90 | **1.49** |
| | + Sub | 38.02 | 33.42 | **2.07** | 37.19 | 37.65 | **0.50** | 40.62 | 29.67 | **5.21** | 39.96 | 37.72 | **3.02** | 37.97 | 41.54 | **1.27** | 35.82 | 33.62 | **4.48** |
| PLLaVA-34B | - | 61.98 | 71.68 | **14.29** | 72.86 | 70.00 | **8.54** | 80.21 | 70.33 | **16.67** | 72.79 | 70.47 | **21.55** | 72.15 | 66.15 | **5.06** | 68.66 | 70.69 | **21.64** |
| | + Sub | 73.50 | 83.16 | **33.87** | 82.41 | 83.53 | **32.16** | 82.29 | 84.62 | **50.00** | 80.78 | 70.97 | **33.19** | 87.34 | 72.31 | **20.25** | 78.36 | 78.45 | **31.34** |
| InternVL2-8B | - | 52.53 | 56.38 | **9.22** | 63.32 | 61.18 | **7.04** | 73.96 | 59.34 | **15.62** | 66.95 | 60.05 | **13.36** | 58.23 | 66.15 | **7.59** | 64.93 | 53.45 | **14.93** |
| | + Sub | 64.52 | 71.68 | **22.12** | 76.38 | 79.41 | **25.13** | 80.21 | 68.13 | **40.62** | 78.62 | 62.28 | **26.72** | 73.42 | 67.69 | **15.19** | 74.63 | 62.07 | **23.13** |
| InternVL2-26B | - | 53.23 | 56.38 | **9.68** | 60.30 | 60.59 | **7.54** | 69.79 | 46.16 | **8.33** | 65.01 | 59.55 | **13.36** | 59.49 | 58.46 | **2.53** | 55.97 | 58.62 | **14.93** |
| | + Sub | 66.59 | 76.28 | **28.11** | 72.86 | 77.65 | **22.11** | 83.33 | 74.73 | **39.58** | 79.91 | 64.52 | **28.23** | 79.75 | 64.62 | **8.86** | 73.88 | 68.97 | **28.36** |
| InternVL2-76B | - | 61.75 | 69.90 | **15.67** | 74.37 | 73.53 | **10.05** | 75.00 | 62.64 | **14.58** | 73.00 | 62.53 | **14.01** | 72.15 | 67.69 | **8.86** | 65.67 | 62.07 | **14.93** |
| | + Sub | 73.50 | 79.85 | **36.18** | 85.43 | 88.24 | **35.68** | 84.38 | 86.81 | **51.04** | 82.51 | 66.75 | **36.21** | 81.01 | 70.77 | **25.32** | 79.85 | 75.86 | **30.60** |
| GPT-4o mini | - | 66.59 | 71.68 | **16.59** | 80.90 | 72.35 | **16.08** | 86.46 | 64.84 | **27.08** | 76.46 | 68.24 | **22.41** | 72.15 | 72.31 | **6.33** | 66.42 | 62.93 | **20.15** |
| | + Sub | 66.59 | 72.19 | **16.82** | 80.90 | 72.35 | **16.08** | 86.46 | 64.84 | **27.08** | 76.46 | 68.24 | **22.41** | 72.15 | 72.31 | **6.33** | 66.42 | 62.93 | **20.15** |
| Gemini 1.5 flash (frames) | - | 67.74 | 73.47 | **21.20** | 78.89 | 75.29 | **22.11** | 79.17 | 72.53 | **31.25** | 74.51 | 65.76 | **24.78** | 75.95 | 73.85 | **13.92** | 70.90 | 67.24 | **26.87** |
| | + Sub | 72.35 | 78.57 | **31.57** | 78.39 | 83.53 | **34.17** | 87.50 | 81.32 | **50.00** | 77.97 | 67.25 | **33.41** | 75.95 | 70.77 | **13.92** | 78.36 | 72.41 | **35.07** |
| Gemini 1.5 pro (frames) | - | 72.35 | 73.21 | **23.96** | 79.40 | 81.18 | **18.59** | 79.17 | 75.82 | **33.33** | 76.67 | 71.71 | **25.86** | 81.01 | 70.77 | **11.39** | 78.36 | 69.83 | **26.12** |
| | + Sub | 79.26 | 82.65 | **44.70** | 84.92 | 88.24 | **45.23** | 85.42 | 87.91 | **58.33** | 82.72 | 70.97 | **42.46** | 81.01 | 70.77 | **27.85** | 78.36 | 75.00 | **45.52** |
| Gemini 1.5 flash (video) | - | 64.06 | 67.35 | **20.51** | 69.85 | 73.53 | **20.60** | 77.08 | 68.13 | **34.38** | 74.30 | 60.55 | **25.22** | 75.95 | 70.77 | **16.46** | 70.15 | 59.48 | **18.66** |
| | + Sub | 74.42 | 77.81 | **32.26** | 79.90 | 82.35 | **27.14** | 86.46 | 76.92 | **43.75** | 80.13 | 64.76 | **29.96** | 73.42 | 64.62 | **11.39** | 82.09 | 75.86 | **35.82** |
| Gemini 1.5 pro (video) | - | 72.58 | 83.16 | **34.79** | 81.91 | 87.06 | **35.68** | 89.58 | 81.32 | **59.38** | 85.31 | 77.42 | **42.24** | 84.81 | 76.92 | **25.32** | 76.12 | 74.14 | **36.57** |
| | + Sub | 76.50 | 80.10 | **41.94** | 79.90 | 84.12 | **35.68** | 86.46 | 84.62 | **50.00** | 83.37 | 71.46 | **43.53** | 79.75 | 78.46 | **30.38** | 79.85 | 73.28 | **43.28** |
| GPT-4o | - | 81.34 | 85.97 | **47.93** | 87.94 | 90.00 | **52.76** | 89.58 | 89.01 | **64.58** | 87.47 | 76.67 | **46.98** | 88.61 | 76.92 | **31.65** | 87.31 | 78.45 | **41.79** |
| | + Sub | 81.34 | 86.48 | **49.08** | 88.44 | 90.59 | **53.77** | 89.58 | 89.01 | **64.58** | 87.26 | 77.42 | **48.28** | 89.87 | 75.38 | **31.65** | 88.81 | 78.45 | **42.54** |
| Gemini 2.5 pro (frames) | - | 82.03 | 82.14 | **43.09** | 86.43 | 85.29 | **34.17** | 86.46 | 87.91 | **60.42** | 87.69 | 78.16 | **44.83** | 88.61 | 80.00 | **29.11** | 88.81 | 78.45 | **50.75** |
| | + Sub | 86.41 | 88.78 | **58.53** | 89.95 | 94.12 | **61.31** | 89.58 | 96.70 | **76.04** | 88.55 | 79.40 | **55.17** | 88.61 | 80.00 | **48.10** | 90.30 | 84.48 | **62.69** |
| Qwen2-VL-7B | - | 63.82 | 76.53 | **17.51** | 75.38 | 75.29 | **16.08** | 82.29 | 62.64 | **19.79** | 76.46 | 67.74 | **21.77** | 74.68 | 72.31 | **15.19** | 72.39 | 72.41 | **26.87** |
| | + Sub | 71.43 | 84.69 | **31.11** | 82.41 | 84.71 | **31.66** | 93.75 | 79.12 | **55.21** | 80.99 | 71.71 | **34.91** | 77.22 | 73.85 | **18.99** | 84.33 | 75.86 | **35.82** |
| | + Ours-FT (RLFT) | 88.71 | 91.58 | **54.84** | 96.48 | 90.59 | **56.28** | 95.83 | 89.01 | **68.75** | 90.06 | 83.87 | **57.76** | 94.94 | 90.77 | **43.04** | 90.30 | 80.17 | **50.00** |
| | + Ours-FT (SFT) | 90.09 | 90.82 | **58.29** | 96.98 | 92.35 | **63.82** | 96.88 | 86.81 | **71.88** | 91.14 | 85.61 | **60.99** | 98.73 | 89.23 | **51.90** | 94.78 | 86.21 | **62.69** |

complexity that leads to a substantial decline in model performance. This further underscores the current limitations of LVLMs in inferring mental states.

## A.4 MORE DETAILS OF R$^3$-BENCH

### A.4.1 VALIDITY OF ANNOTATED CAUSAL CHAINS

In our dataset, each causal chain is approved by three people: the participant who designed the QA pair and two experts for verification (one of whom annotated the causal chain). The participant submits the QA pair and the textual description of the reasoning process behind it, establishing the main content of the causal chain. After being verified by two experts, the filtered data is approved by all three people. The expert in charge of annotation only appropriately expanded and completed the reasoning process without affecting the essence of the causal chain. Also, the annotated causal chains are verified by the other expert. If they fail to pass the verification, the two experts will consult and revise them together. Therefore, we can consider the annotated causal chains to be widely acceptable.

### A.4.2 STATISTICAL ANALYSIS

**Video Statistics.** R$^3$-Bench contains 312 videos. Figure 7a shows the distribution of video durations. All videos are no longer than 180 seconds, with an average duration of 65.8 seconds. The increased video length amplifies the challenge of social reasoning, as identifying relevant social cues becomes more difficult, and causal chains may become longer with more steps, involving a greater variety of events and mental states and requiring tracking of all dynamic changes of all states throughout the video.

**Causal Chain Statistics.** Table 6 presents the statistics of causal chains in our R$^3$-Bench-DX. It includes 347 causal chains composed of 2198 nodes and 1406 single-step subchains. These nodes are categorized into 997 *Event* nodes, 321 *Belief* nodes, 361 *Intent* nodes, 42 *Desire* nodes, and 477

| Videos | R³-Bench-DX | | | | | R³-Bench-Hard |
|---|---|---|---|---|---|---|
| | *EU* | *MSE* | *CW* | *CH/W* | Overall | |
| 312 | 997 | 1201 | 1405 | 1237 | 4840 | 316 |

| Chains | Subchains | Nodes | | | | | |
|---|---|---|---|---|---|---|---|
| | | *Event* | *Belief* | *Intent* | *Desire* | *Emotion* | Overall |
| 347 | 1406 | 997 | 321 | 361 | 42 | 477 | 2198 |

Table 6: Statistics of R³-Bench

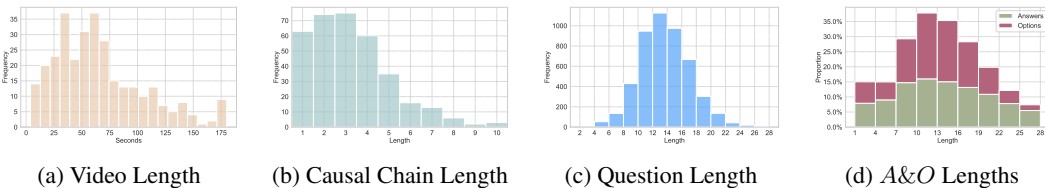

(a) Video Length    (b) Causal Chain Length    (c) Question Length    (d) *A&O* Lengths

Figure 7: Statistics of R³-Bench. *A&O* means *Answer and Incorrect Options*

*Emotion* nodes. Aside from *Desire*, each mental state category has over 300 nodes, providing ample cases to assess specific mental states. In total, 1201 mental state nodes indicate a rich presence of mental state dynamics. The lengths of causal chains range from 1 to 10 steps, with an average length of 3.3 steps, as shown in Figure 7b. Over 60% have no less than three steps, suggesting that the videos require complex and in-depth reasoning.

**QA Statistics.** As detailed in Table 6, there are 4,840 QA pairs in R³-Bench-DX, and there are 316 QA pairs in R³-Bench-Hard. Figure 7c and Figure 7d show the distributions of question, answer, and incorrect option lengths among R³-Bench-DX. The average question length is 13.2 words—longer than many popular VideoQA datasets. Answers and incorrect options average 14.4 and 12.8 words respectively, making the distractors similar in length to the correct answers and thus more challenging.

## A.5 More Details of R³-FDT

### A.5.1 More Details of Information Alignment

The algorithm to extract annotations from existing movie datasets is shown in Algorithm 1. We align the dialogues recognized by Whisper with the time-stamped annotations provided in MovieNet to identify the temporal boundaries of each clip within the full movie. Using these identified temporal boundaries, we then extract the relevant data from both MovieNet and MovieQA, including question–answer pairs, action tags, place tags, and scene descriptions.

---

**Algorithm 1** *ExtractAnnotation*

---

**Require:** Video $v$, Recognized Dialogue $D$, MovieNet Annotation File $F$, MovieQA Set $Q$
**Ensure:** Clip time $(t_s, t_e)$, MovieNet Annotation $M$, MovieQA Annotation $Y$
1: $S_f \leftarrow \texttt{ExtractSubtitle}(F, v)$
2: $S_d \leftarrow \texttt{ExtractDialogue}(D, v)$
3: $E_f \leftarrow \texttt{GetEmbeddings}(S_f)$
4: $E_d \leftarrow \texttt{GetEmbeddings}(S_d)$
5: $(i_s, i_e) \leftarrow \texttt{MatchIndex}(E_f, E_d)$
6: **if** $i_s = $ None **or** $i_e = $ None **then**
7:     **return** None
8: **end if**
9: $(t_s, t_e) \leftarrow \texttt{GetTimestamps}(S_f, i_s, i_e)$
10: $M \leftarrow \texttt{MatchMovieNet}(F, t_s, t_e)$
11: $Y \leftarrow \texttt{MatchMovieQA}(Q, t_s, t_e)$
12: **return** $(t_s, t_e), M, Y$

---

| Videos | EU | Belief | Desire | Intent | Emotion | CW | Overall |
|--------|------|--------|--------|--------|---------|-------|---------|
| 2812 | 14749 | 5645 | 220 | 1552 | 4532 | 14831 | 41529 |

Table 7: Statistics of $R^3$-FDT

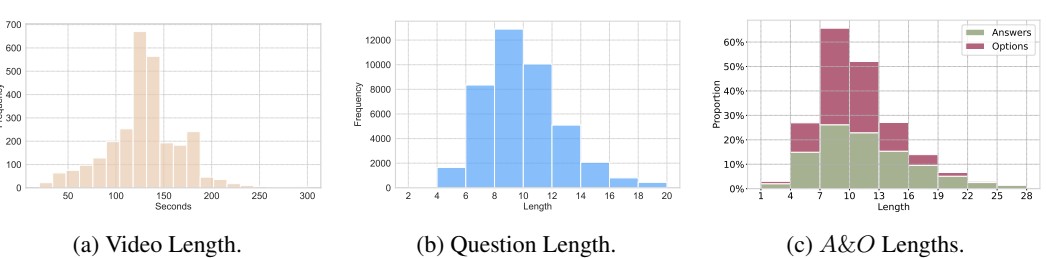

(a) Video Length.  (b) Question Length.  (c) A&O Lengths.

Figure 8: Statistics of $R^3$-FDT. A&O means *Answer and Incorrect Options*

Movie scripts include content beyond the target clips. To avoid using such unrelated information, we extract script segments based on aligned dialogues. As shown in Algorithm 2, we first identify character names as anchors to locate dialogue in the script. We then parse the script $S$ into two parts: scene descriptions and dialogues, yielding a structured script $S'$. Finally, we match the dialogues recognized by Whisper with those in $S'$ and extract the corresponding script segments aligned with each movie clip.

---

**Algorithm 2** *ExtractScript*

---

**Require:** Script $S$, Movie ID $I$, Recognized Dialogue $D$
**Ensure:** Matched Script $S_c$
 1: $C \leftarrow \texttt{GetCharacters}(I)$
 2: $S' \leftarrow \texttt{ParseScript}(S, C)$
 3: $S_c \leftarrow \texttt{Grounding}(S', D)$
 4: **if** $\texttt{Len}(S_c) < \epsilon$ **then**
 5:     **return** $S_c$
 6: **end if**
 7: **return** $\texttt{None}$

---

### A.5.2 STATISTICAL ANALYSIS

**Video Statistics.** Our $R^3$-FDT includes 2812 videos. The average video duration is 128.76 seconds and the median duration is 131 seconds. Figure 8a shows the distribution of video durations.

**QA Statistics.** Figure 8b and Figure 8c show the distributions of question, answer, and incorrect option lengths of $R^3$-FDT. The average question length is 9.55, the average answer length is 11.32 and the average length of incorrect options is 9.88.

### A.6 MORE DETAILS OF EXPERIMENTS

### A.6.1 EVALUATION CONFIGURATION DETAILS

Our evaluation is based on VLMEval (Duan et al., 2024) repository, an open-source toolkit for assessing LVLMs across multiple benchmarks without extensive data preparation. By extending VLMEval, we incorporates our dataset as a new benchmark.

Our task is formulated as a multiple-choice VideoQA problem. Models receive the raw video or selected frames, along with a question and five options, and must select the correct option. We enforce output formatting constraints and determine selected option through exact matching, ensuring reproducible evaluation results. In Section 4.1, we present our specific evaluation metrics.

A.6.2   GRPO TRAINING DETAILS

We implement GRPO training based on verl (Sheng et al., 2024). During training, we input 8 frames to Qwen2-VL-7B, each with a resolution of $280 \times 280$, and input the dialogue recognized by Whisper to the model. We use a rule-based reward function as following:

$$r_f = \begin{cases} 1 & \text{if the response meets the format requirements} \\ -1 & \text{otherwise} \end{cases} \tag{4}$$

$$r_{acc} = \begin{cases} 2 & \text{if the answer is correct} \\ -2 & \text{otherwise} \end{cases} \tag{5}$$

$$r = r_f + r_{acc} \tag{6}$$

We denote the reference model as $\pi_{ref}$, the old policy model as $\pi_{\theta_{old}}$ and the policy model as $\pi_\theta$.

For each question $q$, we sample a group of responses $\mathbf{o} = \{o_1, o_2, ..., o_N\}$ and compute a group of rewards $\mathbf{r} = \{r_1, r_2, ..., r_N\}$. For each $r_i \in \mathbf{r}$, the corresponding advantage $A_i$ is computed as:

$$A_i = \frac{r_i - mean(\mathbf{r})}{std(\mathbf{r})} \tag{7}$$

Then the policy model $\pi_\theta$ is updated according to the following optimization objective:

$$\mathcal{J}_{GRPO}(\theta) = \mathbb{E}\left[q \sim P(Q), \{o_i\}_{i=1}^N \sim \pi_{\theta_{old}}(O \mid q)\right]$$

$$\frac{1}{N}\sum_{i=1}^N \left(\min\left(\frac{\pi_\theta(o_i \mid q)}{\pi_{\theta_{old}}(o_i \mid q)}A_i, \text{clip}\left(\frac{\pi_\theta(o_i \mid q)}{\pi_{\theta_{old}}(o_i \mid q)}, 1-\varepsilon, 1+\varepsilon\right)A_i\right) - \beta\mathbb{D}_{KL}(\pi_\theta||\pi_{ref})\right),$$

$$\mathbb{D}_{KL}(\pi_\theta||\pi_{ref}) = \frac{\pi_{ref}(o_i \mid q)}{\pi_\theta(o_i \mid q)} - \log\frac{\pi_{ref}(o_i \mid q)}{\pi_\theta(o_i \mid q)} - 1, \tag{8}$$

Both the video context and the dialogue are required to answer R³-Bench-DX and R³-Bench-Hard and questions in Social-IQ 2.0 correctly. Therefore, when evaluating on these datasets, we also provide the dialogue to Qwen-2-VL-7B. Previous studies mostly input only the video context to models. When evaluating on IntentQA , we retain the marker for the dialogue section within the prompt but set the dialogue content to "not provided", which is consistent with its "+Sub" setting. We provide the model with a $16 \times 360 \times 640$ video context when evaluating. We report more training details and visualizations in the supplementary material.

A.6.3   SFT DETAILS

We utilize SWIFT (Zhao et al., 2024) to implement SFT and train Qwen2-VL-7B using the same 13k subset. We convert 10% of them into an open-ended format (i.e., removing the options and requiring direct answers). We provide 8 video frames with the model, with each frame having a maximum total pixel resolution of $280 \times 280$. The evaluation settings for this part are identical to those used for the RL fine-tuned model.

A.6.4   DETAILS OF HUMAN STUDY

For R³-Bench-Hard, we invite human subjects to answer all questions.
For R³-Bench-DX, we sample QA pairs corresponding to 43 causal chains for human study. QA pairs derived from the same causal chain are interrelated. To avoid potential shortcuts caused by human memory, we divide all QA pairs corresponding to a single chain into multiple groups based on the following principles: (i) All node QA pairs constitute one group. (ii) For causal QA pairs: Within a group, the node corresponding to the "correct option" of each QA pair must be distinct from the nodes corresponding to the "question" and "correct option" of any other QA pairs in the same group. These principles ensure that no shortcuts exist within the same group. For a single chain, a human subject is only allowed to answer one group of QA pairs.

### A.6.5 COMPUTE RESOURCES

All evaluations were conducted on machines equipped with NVIDIA A100 GPUs. For models with fewer than 10B parameters (e.g., Idefics2-8B), we used a single A100 GPU. For mid-scale models in the 20–40B range, such as PLLaVA-34B and InternVL2-26B, we used two A100 GPUs. Large-scale models with over 70B parameters, including InternVL2-76B, were evaluated using four A100 GPUs. During the training stage, all models were trained using four NVIDIA A100 GPUs.

