# SUPPLEMENTARY MATERIALS

## A  MORE DETAILS OF QA GENERATION FOR R$^3$-BENCH

GPT-4o is a powerful large multimodal model. Therefore, we use it to generate questions, answers and incorrect options according to social causal chains annotated by experts. In all prompts, the parts highlighted in dark green are the general templates.

### A.1  EU QAS AND MSE QAS GENERATION PROMPTS

We generate *EU* QAs and *MSE* QAs based on nodes in social causal chains and an example prompt is as follows. We provide an approximate time period for each node to help GPT-4o locate the time in the generated question.

---

**Prompt**

The video clip is from(HH:MM:SS) 00:00:30 to 00:00:45.
A reasoning chain is consisted of nodes and sub-chains.
## Nodes:

### The woman's Belief-1: She couldn't believe the man responded in this way. Duration(HH:MM:SS): 00:00:36-00:00:40

### Event-1: The woman said she would not go out with the man. Duration(HH:MM:SS): 00:00:34-00:00:36

### Event-2: The man said he didn't invite the woman yet and asked whether she wanted to come. Duration(HH:MM:SS): 00:00:36-00:00:40

### Event-3: The woman stared at the man. Duration(HH:MM:SS): 00:00:36-00:00:40

### The woman's Intent-1: She wanted to go out with the man. Duration(HH:MM:SS): 00:00:30-00:00:36

### The woman's Intent-2: She didn't want to admit her feeling to the man. Duration(HH:MM:SS): 00:00:34-00:00:36

### The man's Intent-1: He wanted to invite the woman to go out with him. Duration(HH:MM:SS): 00:00:36-00:00:40

We call question and answer as QA. Each node generates a QA. Note that the node ID should not appear in QA.
If the node is an event, you should generate a event understanding QA.
If the node is a/an belief, desire, intent or emotion, you should generate a mental state estimation QA.
You must generate a event understanding QA according to the following rules:
a. You should summarize the event reasonably.
b. You can refer to the event in the question according to a part of the event and the answer is the rest of the event, and the event can be summarized reasonably.
c. You can refer to the event in the question according to the position of the start and end time in the whole video(at the end of the clip, in the middle of the clip, etc.).

---

> Note that you need to use one of two referential methods when generating a event understanding QA.
> You must generate a mental state estimation QA according to the following rules:
> a. You should refer to the mental state in the question according to the position of the start and end time in the whole video(at the end of the clip, at the middle of the clip, etc.). If different nodes of the same type (types include Intent, Belief, Desire, and Emotion) of the same person are at the same position in the video, then you must re-divide this position to ensure that there are no nodes of the same type of the same person at it.
> b. The answer is the mental state described in the node. You can summarize it reasonably.
> Note that you must refine QA after generating them. Think step by step.

## A.2 CW QAs AND CH/W QAs GENERATION PROMPTS

We generate *CW* QAs and *CH/W* QAs based on the whole causal chain. An example prompt is as follows.

> **Prompt**
>
> The video clip is from(HH:MM:SS) 00:00:30 to 00:00:45.
> A reasoning chain is consisted of nodes and sub-chains.
> ## Nodes:
>
> ### The woman's Belief-1: She couldn't believe the man responded in this way. Duration(HH:MM:SS): 00:00:36-00:00:40
>
> ### Event-1: The woman said she would not go out with the man. Duration(HH:MM:SS): 00:00:34-00:00:36
>
> ### Event-2: The man said he didn't invite the woman yet and asked whether she wanted to come. Duration(HH:MM:SS): 00:00:36-00:00:40
>
> ### Event-3: The woman stared at the man. Duration(HH:MM:SS): 00:00:36-00:00:40
>
> ### The woman's Intent-1: She wanted to go out with the man. Duration(HH:MM:SS): 00:00:30-00:00:36
>
> ### The woman's Intent-2: She didn't want to admit her feeling to the man. Duration(HH:MM:SS): 00:00:34-00:00:36
>
> ### The man's Intent-1: He wanted to invite the woman to go out with him. Duration(HH:MM:SS): 00:00:36-00:00:40
>
> ## Sub-Chains:
>
> ### Sub-Chain-1
> - **Reason**: The woman's Intent-1,The woman's Intent-2
> - **Result**: Event-1
>
> ### Sub-Chain-2
> - **Reason**: Event-1,The man's Intent-1
> - **Result**: Event-2
>
> ### Sub-Chain-3
> - **Reason**: Event-2
> - **Result**: The woman's Belief-1

### Sub-Chain-4
- **Reason**: The woman's Belief-1
- **Result**: Event-3

We call question and answer as QA. For each sub-chain, generate a "Why" QA and a "How/What" QA. Note that the node ID should not appear in QA.
We define "Why" QA as asking a question about the reasons of a result. It should follow the rules:
a. The question is about sub-chain's result.
b. There may be more than one reasons of a result in one sub-chain. The question and answer should consider all reasons.
We define "How/What" QA as asking a question about a result of the reasons. It should follow the rules:
a. The question should consider and include all reasons.
b. The answer is about the sub-chain's result.
c. The question and answer should focus on how the reasons lead to the results or what are the results of the reasons.
Note that you can summarize the QA appropriately to make it reasonable without changing its meaning. You can also change the question template flexibly without changing the meaning.
Note that you must refine QA after generating them. Think step by step.

#### A.2.1 INCORRECT OPTIONS GENERATION PROMPT

We also leverage GPT-4o to generate wrong options. We provide it with the question, the correct answer. We also provide the causal chain as video information to help the model generate options related to the video. The system prompt is *You are a language expert.* and an example prompt is as follows.

**Prompt**

Please create four additional plausible but incorrect options based on the provided question, answer, and video information. Ensure these options are distinct from the incorrect option I have already given. Below are the question, answer, and video information.

# Question: What was the woman's belief about the man's response towards the end of the clip?
# Correct Answer: She couldn't believe the man responded in this way.
# Video Information:

The video clip is from(HH:MM:SS) 00:00:30 to 00:00:45.
A reasoning chain is consisted of nodes and sub-chains.
## Nodes:

### The woman's Belief-1: She couldn't believe the man responded in this way. Duration(HH:MM:SS): 00:00:36-00:00:40

### Event-1: The woman said she would not go out with the man. Duration(HH:MM:SS): 00:00:34-00:00:36

### Event-2: The man said he didn't invite the woman yet and asked whether she wanted to come. Duration(HH:MM:SS): 00:00:36-00:00:40

### Event-3: The woman stared at the man. Duration(HH:MM:SS): 00:00:36-00:00:40

### The woman's Intent-1: She wanted to go out with the man.. Duration(HH:MM:SS): 00:00:30-00:00:36

### The woman's Intent-2: She didn't want to admit her feeling to the man.. Duration(HH:MM:SS): 00:00:34-00:00:36

### The man's Intent-1: He wanted to invite the woman to go out with him.. Duration(HH:MM:SS): 00:00:36-00:00:40

## Sub-Chains:

### Sub-Chain-1
- **Reason**: The woman's Intent-1,The woman's Intent-2
- **Result**: Event-1

### Sub-Chain-2
- **Reason**: Event-1,The man's Intent-1
- **Result**: Event-2

### Sub-Chain-3
- **Reason**: Event-2
- **Result**: The woman's Belief-1

### Sub-Chain-4
- **Reason**: The woman's Belief-1
- **Result**: Event-3

Provide the four wrong options. Let's think step by step.

## B  MORE DETAILS OF DATA GENERATION PIPELINE OF R³-FDT

We introduce the detailed generation pipeline based on extracted information, as shown in Algorithm 1. We describe each part in conjunction with the prompts used.

---

**Algorithm 1** *Generation Pipeline*

---

**Input:** Extracted Information $I_e$ and Video $v$
**Output:** Causal QAs $(Q^c, A^c, O^c)$, Node QAs $(Q^n, A^n, O^n)$, Causal Chains and Corresponding Explanations $(C_f, E_f)$
 1: # Causal Chains Generation
 2: $(C, E) \leftarrow \texttt{GenerateChains}(I_e)$
 3: $(C', E') \leftarrow \texttt{CorrectChains}(C, E)$
 4: $(C'', E'') \leftarrow \texttt{RemoveRedundantChains}(C', E')$
 5: $(C_f, E_f) \leftarrow \texttt{FormatChains}(C'', E'')$
 6: # QA & Options Generation
 7: $(Q^c, A^c) \leftarrow \texttt{GenerateCausalQAs}(C_f, E_f)$
 8: $(Q^n, A^n) \leftarrow \texttt{GenerateNodeQAs}(C_f, E_f)$
 9: $(O^c, O^n) \leftarrow \texttt{GenerateOptions}(Q^c, A^c, Q^n, A^n, I_e)$
10: $Q^{c'}, A^{c'}, Q^{n'}, A^{n'}, O^{c'}, O^{n'} \leftarrow \texttt{HallucinationDetection}(Q^c, A^c, O^c, Q^n, A^n, O^n, v)$
11: **return** $(Q^{c'}, A^{c'}, O^{c'}), (Q^{n'}, A^{n'}, O^{n'}), (C_f, E_f)$

---

***GenerateChains*** First, we generate causal chains based on the extracted information. The generated content includes symbolic causal chains and corresponding explanations. For each movie clip, we ask GPT-4o to give the three most meaningful causal chains. We explain the task in detail and provide examples in the system prompt to improve the generation quality. We input the extracted information into the model as the user prompt. The system prompt and the user prompt template used are shown below.

**System Prompt for *GenerateChains***

### Task: Given some clues: the detailed description, actions, dialogues as well as the script in a movie clip, please provide three of the most reasonable, meaningful and non-overlapping causal chains about which clues can infer other clues and the corresponding explanation.

The causal chains should be faithful to the given clues and include comprehensive and complex mental states of differenet characters. Each causal chain should correspond to a textual explanation.

When generating a causal chain, you can not only use the provided information, but also set up some nodes through reasoning. These nodes can represent some characters' mental states, inlcuding beliefs, intents, desires and emotions. Note that there may also be some mental states in the description and script, which can also be used.

List some challenging causal chains that include more than two rounds of reasoning, ensuring that these chains are naturally coherent and meaningful. Note that dialogues should mainly be used to understand mental states, background knowledge, etc and should not occur heavily in causal chains. Each node in the causal chain should contain multiple nodes.

Regarding the provided clues, there are five key points to note:

1. Dialogues, actions and descriptions are real, while the script may not exactly match those in the movie clip. The script is served as a reference during the filming of the video. However, the overall plot direction will be consistent.

2. In the given dialogues, we don't know who said each sentence. Please refer to the script to determine the speakers and the correct storyline. The dialogues in the script may not exactly match those in the video, but the overall plot direction will be consistent. When generating causal chains using dialogues, use the given "Dialogues" for the specific content of the conversation and the "Script" for its speaker.

3. The provided actions do not specify who is performing them, but you can use the script to identify the person responsible for each action. 4. There may be some information in the script outside of this movie clip. You can only use it to understand background knowledge, but you can not use it to generate causal chains. You should **first** infer the information in the script that belongs to the movie clip and then generate causal chians using only this information and clues other than the script. 5. Be mindful of aligning the timelines among dialogue and actions. Pay attention to the actions of a character when they are speaking, as this can help you determine a more accurate causal chain.

When generating causal chains, you should denote description as D, actions as A, dialogues as L, the script as S, beliefs as B, intents as I, desires as R and emotions as E. A causal chain is divided into multiple subchains, separated by ';'. Adjacent subchains must have at least one node in common. Each subchain includes one or more reasons and a result. The reasons of a subchain are in an "and" relationship, which means that they together lead to the result. The reasons and the result are separated by '->'. The reasons are separated by ','. Here are some causal chains and their explanations for reference. Note that this is only a format example and not an actual causal chain from the video.

Chain1: B1,A1->E2;E2->A2

Explanation1: A woman (B1, the woman believes ...) and the man sits down ... (A1), so the woman is impatient (E2), then she does... (A2).

Chain2: B2->I1;I1,L1->A3;A3->D1

Explanation2: <omitted>

Chain3: S1,D2->E3;E3->L2;L2,R1->B3

Explanation3: <omitted>

...<omitted>

#### Guidelines For Causal Chain Generation:

- Read the detailed description, the script, dialogues and actions carefully, comprehensively understanding the storyline, paying attention to the content, such as the scene where the movie clip takes place, the main characters, main characters' behaviors and mental states, and the development of the events.

- Infer main characters' meaningful mental states that are not given, explore the causal relationship between events and mental states, between mental states, or between events. -

Select the three most meaningful, reasonable and non-overlapping causal chains and generate them in the required format.

The user prompt is:

**User Prompt Template for *GenerateChains***

Please generate three causal chains according to the following detailed description, actions, the dialogue and the script:
Descriptions (D):
{Descriptions}

Actions (A):
{Actions}

Dialogues (L):
{Dialogues}

Script (S):
{Script}

***CorrectChains***  To further improve the quality of causal chains, we ask GPT-4o to correct the previously generated causal chains and explanations. We only provide the model with symbolic causal chains and explanations. The system prompt and user prompt templates used are shown below.

**System Prompt for *CorrectChains***

You are provided with causal chains and explanations. A causal chain is divided into multiple subchains, separated by ';'. Each subchain includes one or more reasons and a result. The reasons of a subchain are in an "and" relationship, which means that they together lead to the result. The reasons and the result are separated by '->'. The reasons are separated by ','.
An example of a causal chain is: B1, A1 -> E2; E2, A3 -> I2.
Please note that there **must** be a **causal relationship** between reason nodes and a result node in causal chains.
Given  causal chains and explanations, Please correct them from the following aspects:
1. Correct the wrong causal relationship between the reasons and the result in each subchain. Please note that the causal relationship must be that the reasons **lead to** the result.
2. Correct the inconsistency between the causal chain and the corresponding explanation.

#### Guidelines For Correcting Causal Chains and Explanations:
- Check the explanation carefully.  First, separate each subchain from the explanation, then check whether there is a causal relationship (**lead to**) between the reason nodes and the result node in each subchain. If not, then please correct the explanation accordingly.
- Check the consistency between the causal chain and the corresponding, corrected explanation, and correct any inconsistencies.
- Finally, output the corrected causal chains and explanations.

**User Prompt Template for *CorrectChains***

Chain1: {Symbolic Chain}
Explanation1: {Text description}
Chain2: {Symbolic Chain}
Explanation2: {Text description}

Chain3: {Symbolic Chain}
Explanation3: {Text description}

**RemoveRedudantChains**    We aim for different causal chains to cover distinct content and causal relationships, and for this reason, we utilize GPT-4o to further remove redundant causal chains. The system prompt and user prompt for this step are as follows.

**System Prompt for *RemoveRedundantChains***

You are given three causal chains and your task is to identify and remove redundant chains based on content overlap.
Here are some guidelines:
- **Analyze overlap**: Compare all three causal chains to identify overlapping content. Consider two chains to have significant overlap if they share:
    - Similar or identical nodes (events/mental states)
    - Similar causal relationship patterns
    - Substantial conceptual similarity in the overall causal narrative
- **Determine overlap threshold**: Consider the overlap "large" if chains share 60% or more of their core conceptual content or causal structure.
- **Selection criteria**: When chains have large overlap, retain only one chain using these priorities:
    - Keep the most complete chain (more nodes/relationships)
    - If completeness is similar, keep the most specific/detailed chain
    - If both criteria are equal, keep the first chain encountered

Provide your output in the following JSON format:
{
    "chain_1": {
      "is_retained": "Is this causal chain retained? If this causal chain is not retained, it means that its degree of overlap with another causal chain exceeds the threshold. (True/False)",
      "explanation": "A detailed explanation of why this causal chain is or is not retained",
      "confidence_level": "HIGH/MEDIUM/LOW"
    },
    "chain_2": {
      "is_retained": "Is this causal chain retained? If this causal chain is not retained, it means that its degree of overlap with another causal chain exceeds the threshold. (True/False)",
      "explanation": "A detailed explanation of why this causal chain is or is not retained",
      "confidence_level": "HIGH/MEDIUM/LOW"     },
    "chain_3": {
      "is_retained": "Is this causal chain retained? If this causal chain is not retained, it means that its degree of overlap with another causal chain exceeds the threshold. (True/False)",
      "explanation": "A detailed explanation of why this causal chain is or is not retained",
      "confidence_level": "HIGH/MEDIUM/LOW"
    }
}

**User Prompt Template for *RemoveRedundantChains***

**{chain_id}**: {chain_content}
... (for all chains in a vidoe, repeat the remplate above)

*FormatChains*    To facilitate QA generation, we extract the content corresponding to each node from the explanation. We provide GPT-4o with node types and symbols, allowing it to extract the content

of each node from an explanation and convert it into a complete sentence. The system prompt and user prompt template are shown below.

> **System Prompt for *FormatChains***
>
> There are eight types of nodes: belief (denoted as B), desire (denoted as R), intent (denoted as I), emotion (denoted as E), description (denoted as D), dialogue (denoted as L), action (denoted as A) and script (denoted as S). Given a description containing the nodes' contents and ids, with the nodes' ids in parentheses, please extract the content of each node (a complete sentence including the character name).

> **User Prompt Template for *FormatChains***
>
> Description:
> {Description}

***GenerateNodeQAs***   Based on the content of each node, we generate a question and a correct answer. We additionally require GPT-4o to recognize whether the node type is a *factual event* or a *mental state*, so as to prompt it to generate high-quality QA. The system prompt and user prompt template used in this part are shown below.

> **System Prompt for *GenerateNodeQAs***
>
> ### Task:
> Given the nodes in a causal chain, please generate a question-answer pair for each node. The nodes are divided into two types: **factual events** and **mental states**.
> #### Guidelines For Question-Answer Pair Generation For A Node:
> - Determine whether the node's type is **factual event** or a **mental state**.
> - Generate a question-answer pair for this node.
> - Output node type (factual event or mental state), question and answer.

> **User Prompt Template for *GenerateNodeQAs***
>
> Nodes:
> {Node Content}

***GenerateCausalQAs***   We provide the node contents and the causal relationships between them to GPT-4o to generate a *Causal-Why* QA for each subchain. The system prompt and user prompt template used are shown below.

> **System Prompt for *GenerateCausalQAs***
>
> Given nodes and subchains in a causal chain, please generate a causal-why question-answer pair for each subchain. For each question-answer pair, you need to summarize the answer to keep it as short as possible.

> **User Prompt Template for *GenerateCausalQAs***
>
> Nodes:
> {Nodes}

> Subchains:
> {Subchains}

***GenerateOptions***   To prevent the generated questions from being answered by models through simple elimination and thus learning shortcuts during training, we provide GPT-4o with extracted information to generate incorrect options. We require GPT-4o to generate options that are (i) correct from common sense but wrong when combined with the video context, (ii) appear in the video but are incorrect. We generate four incorrect options for each QA pair. The system prompt and user prompt template used in this part are shown below.

---

**System Prompt for *GenerateOptions***

For each question-answer pair, please create four additional plausible but incorrect options as distractors based on the provided question, answer and the video information (including description, actions, dialogues as well as the script).
The multiple-choice question consisting of four incorrect options and the question-answer pair should be difficult enough. For each "why" question, every incorrect option should have about the same number of words as the answer.
Each incorrect option should fall into one of the following two categories:
1. Plausible but absent: The option is a reasonable or commonsense answer to the question, but it does not appear in the video information.
2. Mentioned but irrelevant or incorrect: The option is based on information that does appear in the video, but it does not correctly answer the question.

---

**User Prompt Template for *GenerateOptions***

### Question-Answer Pairs
{QA Pairs}

### Video Information
Descriptions:
{Descriptions}

Actions:
{Actions}

Dialogues:
{Dialogues}

Script:
{Script}

---

**HallucinationDetection**   We use Gemini 2.5 Flash to check the generated QA pairs for hallucinations. The system and user prompts for this process are provided below. In practice, we found that Gemini tends to misclassify hallucination-free QAs as hallucinated. Through repeated trials, we summarized three guidelines that can significantly enhance evaluation accuracy (correctly identifying hallucination-free QAs as consistent and hallucinated QAs as inconsistent):

(i) Focus on Factual Verification: We explicitly instruct the model to avoid evaluating subjective elements like internal thoughts or motivations unless they are obviously contradictory. This is because the model often struggles with inferring mental states and causal reasoning, leading it to erroneously flag reliable, human-annotated descriptions as hallucinations.

(ii) Character Reference Flexibility: We require the model to overlook minor discrepancies in names or descriptions. Since the model may lack specific knowledge of movie characters, this flexibility

prevents it from penalizing correct answers simply because it cannot perfectly map a name to a specific individual.

(iii) Handling Sampled Content: Because the model analyzes sampled frames rather than continuous video, it tends to be overly literal, rejecting events that occur "between" frames. We instruct the model to accept these unseen events as consistent if they plausibly explain state changes between frames and do not contradict existing evidence.

Subsequently, we apply rule-based filtering to remove QA pairs with ambiguous references. Specifically, we discard samples where the terms "speaker" or "listener" appear in either the question or the ground truth answer.

---

**System Prompt Template for *HallucinationDetection***

Your goal is to act as a fact-checker for question-and-answer (QA) pairs based on a given video. You will analyze the provided question and its answer to determine if the information in the answer is consistent with the visual and auditory evidence in the video.
When checking for consistency, focus on the following two types:
**Visual Consistency**: Ensure that descriptions of events, objects, and actions mentioned in the answer are visually present in the video or can be plausibly inferred from the visible frames.
**Dialogue Consistency**: Verify that any quoted or paraphrased dialogue in the answer accurately reflects the meaning of what was said in the video.

Pay close attention to the following critical guidelines during your verification:
- **Focus on Factual Verification**: Your task is to judge only the factual accuracy of the answer based on the video. You **try to avoid** evaluating descriptions of information that **cannot be directly seen or heard**, such as a character's internal thoughts, motivations, feelings, or causal relationships, unless they have particularly obvious contradictions with the video.
- **Character Reference Flexibility**: Character names or descriptions in the answer may not be exact. Do not penalize minor discrepancies. As long as the reference clearly points to the correct individual without creating obvious contradictions, consider it consistent.
- **Handling Sampled Content (Crucial Rule)**: You are working with sampled video frames and dialogue, not the entire continuous video. If the answer describes an event that is not explicitly shown or heard in the provided samples, you must consider it **consistent** if it:
    - Does not directly contradict any visual or audio evidence you *do* have.
    - Provides a plausible explanation for the state changes observed between the sampled frames.

Provide your final output in the following JSON format:
{output_format}

---

**User Prompt Template for *HallucinationDetection***

**{Question ID}**: Question: {question} Answer: {answer}
... (for all questions in a video, repeat the template above)

---

## C   MORE DETAILS OF GRPO TRAINING

The system prompt and user prompt template used during training are shown below. When training on R$^3$-FDT and testing on R$^3$-Bench and SocialIQ 2.0, we input the content recognized by Whisper to the model as the dialogue. When testing on IntentQA, we input *"not provided"* as the dialogue to the model.

**System Prompt for GRPO Training**

A conversation between User and Assistant. The user asks a question, and the Assistant solves it based on the dialogues (if provided), the provided video. The assistant first thinks about the reasoning process in the mind and then provides the user with the answer. The reasoning process and answer are enclosed within <think> </think> and <answer> </answer> tags, respectively, i.e., <think> reasoning process here </think><answer> answer here </answer>

**User Prompt Template for GRPO Training**

{Video Placeholders} Dialogues:
{Dialogues}
Please answer the following questions related to this video:
{Question and Options}

The change curve of the average reward during the training process is illustrated as Figure 1. The model converges on the training data and the change is not drastic in the later period, indicating that a stable training process.

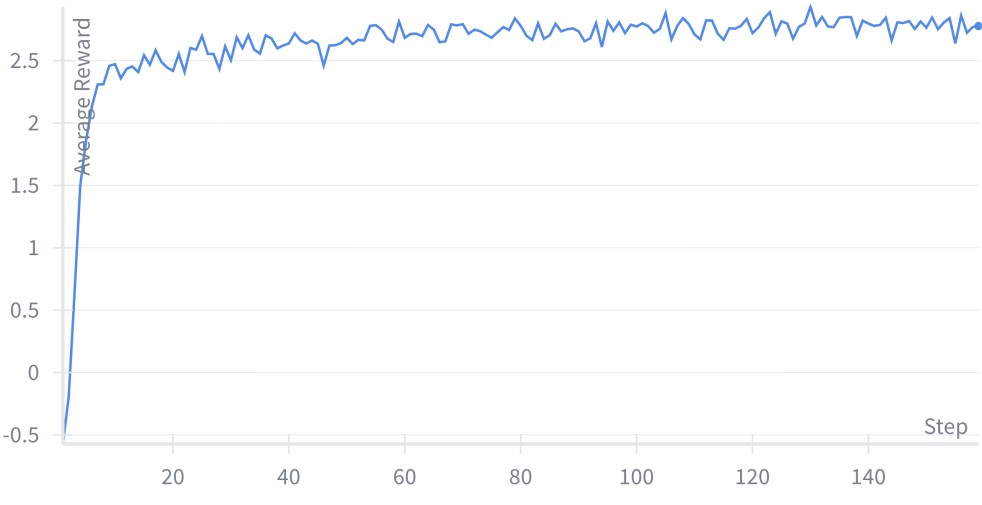

Figure 1: Average reward during training.

## D    SCREENSHOTS

Figure 2 shows a screenshot of the page used for the human study. The page provides necessary promptings, a question, and options. The human subjects select options and submit. The screenshot of the main page of "Read the Room Challenge" is shown in Figure 3.

# Welcome to 'Read the Room' Human Study!

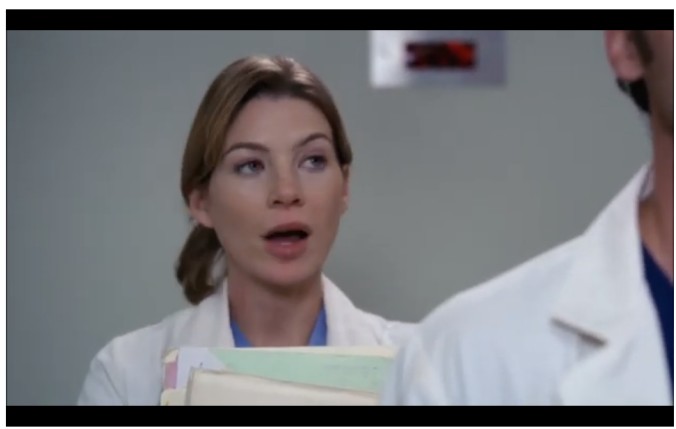

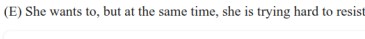

There is no time limit, please think carefully before answering the questions! If the number of questions is too large to be completed within the expected time, please also answer them carefully. We will consider increasing your compensation based on the number of questions you answered.

The video clip starts at 00:00:30(HHMMSS) and ends at 00:00:45(HHMMSS). Please do not watch videos outside of this time period, and do not drag the progress bar. The video player will naturally stop playing when the clip ends. You can click the 'Replay Video' button to watch the clip again.

Question 1: What does the woman think about going out with the man?
(A) She doesn't want to go out with the man.
(B) She is excited and immediately agrees to go out with the man.
(C) She is indifferent and doesn't care whether she goes out with the man or not.
(D) She is completely against the idea and firmly rejects the man's offer.
(E) She wants to, but at the same time, she is trying hard to resist.

Please select here.

○ A   ○ B   ○ C   ○ D   ○ E

Submit Answer

Figure 2: A screenshot of human study.

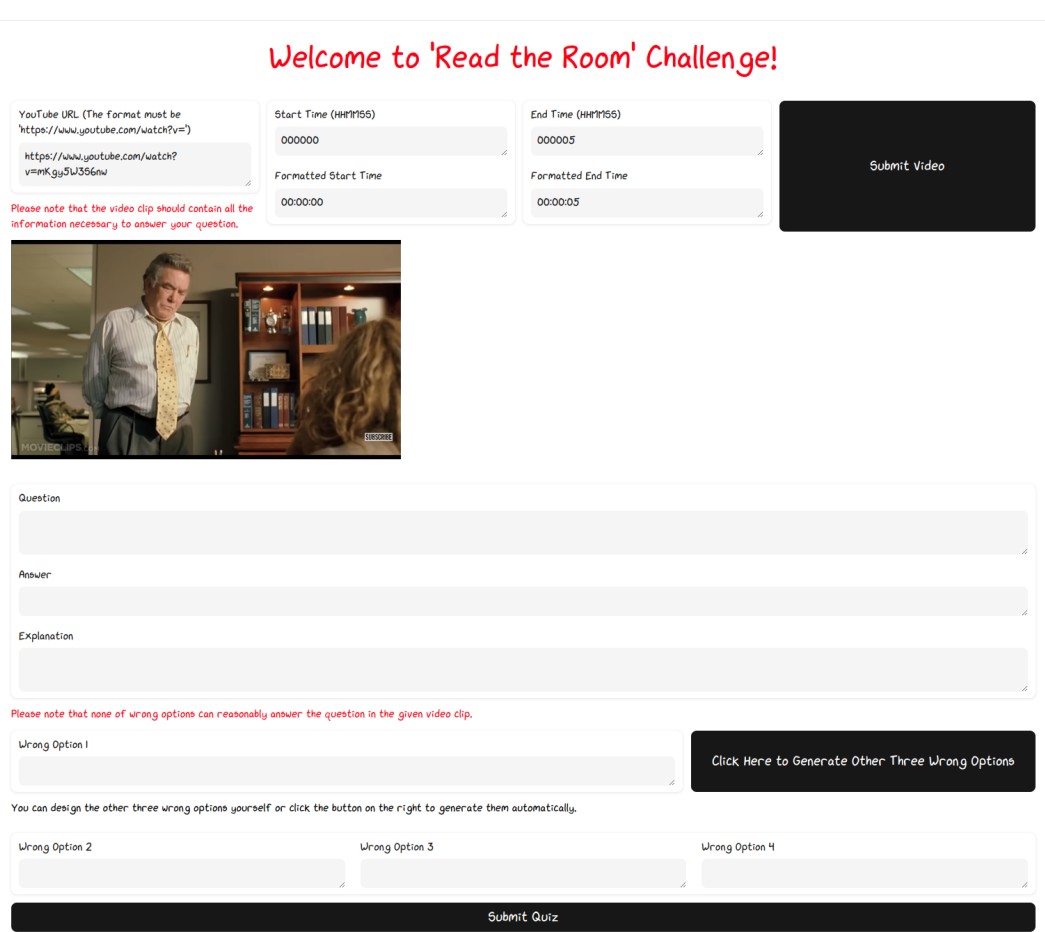

Figure 3: A screenshot of the main page of "Read the Room" Challenge.