# OpenReview forum: "Read the Room: Video Social Reasoning with Mental-Physical Causal Chains"
_ICLR.cc/2026/Conference — ICLR 2026 Poster_

### Official Review · Reviewer_G5Ui · 2025-10-30

**Soundness:** 3
**Presentation:** 3
**Contribution:** 3
**Rating:** 6
**Confidence:** 4

**Summary:**

The authors introduce a new dataset, $R^{3}$-VQA, to address the challenge of teaching AI to infer complex social and mental states from video. The dataset contains two parts: $R^{3}$-Bench, a high-quality video QA benchmark with fine-grained manual annotations for mental states (belief, intent, desire, emotion) and the multi-step causal chains that connect them to physical events; $R^{3}$-FDT, an training dataset augmented from existing datasets using a similar automatic framework. The authors showed that state-of-the-art multimodal LLMs fall short of human performance on $R^{3}$-Bench, and finetuning a model on $R^{3}$-FDT with GRPO yielded improvements across datasets.

**Strengths:**

- $R^{3}$-Bench contains rich, manual annotation of social reasoning chains. It also has decent scale (312 videos, 4,840 QAs) for an evaluation set
- While $R^3$-FDT has no human-in-the-loop verification, using it for RL finetuning yielded significant improvements across datasets (including Social-IQ and IntentQA). The pipeline also seems scalable and adaptable.
- The difficult subset $R^{3}$-Bench-Hard showed significant gap between AI and human accuracy, demonstrating room for improvement.

**Weaknesses:**

- $R^3$-FDT is automatically generated, and the paper lacks crucial details on its verification. A "Hallucination Detection" step is mentioned, but the exact criteria for filtering, the specific prompts used, and the rejection rate (i.e., how many QA pairs were filtered out) are not reported. This makes it difficult to assess the final quality and potential noise level of the training data.
- The main evaluation set, $R^{3}$-Bench-DX, appears to be approaching saturation. The top-performing models, such as the finetuned Qwen2-VL-7B and Gemini 2.5 Pro, are close to human accuracy performance. This limits its long-term utility as a challenging benchmark.
- The "Hard" subset, which shows a much larger performance gap, is very small, containing only 316 questions. This limited scale and diversity make it difficult to rely on as a robust and comprehensive evaluation set for such a complex task.

**Questions:**

- Could the authors please elaborate on the construction of $R^{3}$-Bench-Hard? The paper states it was "sourced from the winning submissions of a social reasoning challenge", but what were the specific selection criteria? What attributes (e.g., causal chain length, type of mental state, presence of deception) make this subset quantifiably "harder" than $R^{3}$-Bench-DX? Given its small size, providing a few qualitative examples that directly compare a "Hard" question to a "DX" question would be beneficial.
- The novel consistency metric is appreciated, but its interpretation is difficult. The criterion -- where a chain is "consistent" only if all associated questions are answered correctly -- seems overly strict and may be the reason for the low consistency scores. The authors omit a human baseline for this metric, citing concerns about human memory. However, a baseline, even if imperfect, would be a good reference point to contextualize the models' low consistency scores. Could the authors provide this consistency score human baseline, or discuss why a modified human study was not feasible?
- The "high accuracy, low consistency" paradox  is ambiguous. The consistency metric doesn't distinguish between two very different issues: 1) A model answers a high-level causal question correctly without correctly answering the preceding, simpler questions in the chain. This would suggest the high-level question is flawed or hackable with a heuristic, not by causal reasoning. 2) A model answers basic event/mental state questions correctly but fails at the more complex, high-level reasoning questions that come later in the chain. This indicates a genuine model failure. Can the authors clarify whether the current metric distinguishes these two cases? A more granular analysis or metric seems necessary to determine if the low consistency is due to flawed questions (Case 1) or genuine model reasoning deficits (Case 2).
- Nit: there's an overlap of Table 1 on page 3

Overall, while I lean towards acceptance due to the dataset's demonstrated utility for fine-tuning, I have the above reservations about the saturation of the evaluation set and whether the benchmark robustly measure genuine social understanding.

---

> ### Author Response · Authors · 2025-11-23
>
> We sincerely appreciate your careful review and valuable feedback. We have corrected the overlap of Table 1 and we will do our best to address your concerns.
> ### Q1: Elaboration on $R^3$-Bench-Hard
> We apologize for the confusion caused by our unclear statement. In fact, **$R^3$-Bench-Hard consists of data verified by experts during the "Human Data Verification" stage in Section 2.2.** Its composition and specific selection criteria are detailed in the "Human Data Collection" part of the same section ((1) to (5)). Since we collected this data via the "Read the Room Challenge" during the "Human Data Collection" stage, we unclearly described it in the original paper as being "directly sourced from the winning submissions of a social reasoning challenge." \
> $R^3$-Bench-DX is the QA data verified by experts in the final "QA Generation & Verification" stage. It consists of samples derived from the causal chains annotated based on the samples in $R^3$-Bench-Hard. We have clearly stated the sources and composition of both $R^3$-Bench-Hard and $R^3$-Bench-DX in **Section 2.2** of the revised paper (highlighted in blue). \
> Regarding the attributes that make the subset "harder": Although we did not provide crowdworkers or experts with prior of "attributes making the questions hard" during data collection and verification, we asked experts to analyze and summarize the data in $R^3$-Bench-Hard. They eventually categorized them into six cognitive dimensions—these dimensions are the key attributes that contribute to the difficulty. Furthermore, because $R^3$-Bench-DX is derived from the nodes and subchains of annotated causal chains—focusing on single cues and single-step causal reasoning—it is simpler than $R^3$-Bench-Hard, which involves multi-step reasoning.  \
> We provide a qualitative example to intuitively illustrate the difference in difficulty between $R^3$-Bench-Hard and $R^3$-Bench-DX. In a video clip, although a woman claims she does not want to go out with a man, her coy expression and the act of answering before being asked clearly indicate that she actually wants to go out with him.
> - In $R^3$-Bench-Hard: The corresponding question is "What does the woman think about going out with the man?" with the ground truth answer: "She wants to, but at the same time, she is trying hard to resist." This QA requires high-level reasoning that synthesizes the entire social interaction process within the video clip.
> - In $R^3$-Bench-DX: The expert-annotated causal chain details the conversation triggered by the woman's conflicting intentions (wanting to go but not wanting to admit it) and the man's invitation. One specific subchain is: "The woman wanted to go out with the man, but she didn't want to admit her feeling to the man (reasons), so she said she would not go out with the man (result)." This subchain derives a QA regarding the woman's intent (reasons), but it also derives a simple QA (Question: "What did the woman express about going out with the man in the middle of the clip?"; Ground truth answer: "She said she would not go out with the man.") regarding her stated content (result)
>
> These simple facts are necessary for modeling the entire social interaction process. Therefore, deriving such simple QAs is inevitable. This is why $R^3$-Bench-DX is simpler. However, please note that these simple QAs are helpful for the "diagnostic" role of $R^3$-Bench-DX. Although they are simple for closed-source strong models, they are beneficial for diagnosing specific capability deficiencies in open-source models.

---

> ### Author Response · Authors · 2025-11-23
>
> ### Q3: Two cases of low consistency.
> Thanks for your excellent question. Broadly speaking, both Case 1 and Case 2 represent scenarios where the model fails to "truly understand" causal relationships. While our consistency metrics effectively determine "whether the model truly understands causal relationships"—a capability lacking in existing datasets—the metric itself acts as a high-level indicator and does not inherently distinguish whether the low consistency stems from Case 1 or Case 2. However, please note that this distinction is achievable through causal chains. We can categorize inconsistent subchains into specific cases. By analyzing the proportions of these cases, we can determine the underlying causes of the model's low consistency. \
> Before presenting the experimental results, we wish to clarify a critical point: **Regarding Case 1, high-level causal questions are inherently hackable (please note that this is an intrinsic flaw of the questions themselves, rather than a defect in our dataset). Models can employ heuristics or shortcuts—such as relying on common sense or eliminating options that did not occur in the video—to answer these questions correctly.** From a cognitive perspective, models should ideally comprehend **basic event/mental state questions** before inferring **high-level causal relationships**. However, in practice, we find that models often exhibit the reverse behavior: they appear to understand causal relationships better than events and mental states. For instance, IntentQA reports that model performance on CW and CH (causal questions) is higher than on TP and TN, providing strong evidence. However, since existing datasets lack detailed annotated causal chains, they fail to detect this phenomenon. This aligns perfectly with the **Case 1** scenario you mentioned. For example, consider a causal question from NextQA and IntentQA: "why did the short hair girl stand up on the piano chair after playing the piano for awhile?\nA. even out cake\nB. to slide from left to right\nC. prevent her from eating\nD. get something from the piano top\nE. make musical sound\n". This question can be answered correctly by merely reading the text without watching the video. \
> **Experiments**: Leveraging causal chains, we can categorize inconsistent sub-chains into three scenarios: **Case 1**, **Case 2**, and **Case 3** (where the model fails to correctly answer both the **basic event/mental state questions** and the **high-level causal questions**). We analyze the proportions of these three cases to determine the primary causes of low consistency. As shown in Figure 7 in Section A.3.1 of the revised paper, we find that for lower-performance models (e.g., PLLaVA, Video-LLaVA), Case 3 accounts for the largest proportion, indicating severe deficiencies in both fundamental capabilities and high-level reasoning. Conversely, for high-performance models (e.g., Gemini, GPT), the combined proportion of Case 1 and Case 2 increases significantly, with Case 1 being the most dominant. This highlights the prevalence and severity of the aforementioned **"hijacking"** phenomenon. **While other datasets might overlook this issue and report overestimated causal reasoning capabilities, our consistency metric accurately detects it—allowing us to distinguish between flawed/hackable questions (Case 1) and genuine model reasoning deficits (Case 2)—and offers a more reliable assessment**, fully demonstrating the significant value of our evaluation set.
>
> ### W1: Crucial Details of "Hallucination Detection" lack.
> Thank you for pointing it out. We have added the prompt used to **'HallucinationDetection' within Section C ('MORE DETAILS OF DATA GENERATION PIPELINE OF R3-FDT')** of the supplementary material. To ensure robust hallucination detection, we instruct the model to verify **Visual Consistency** (ensuring events indeed occur in the video) and **Dialogue Consistency** (verifying quoted or paraphrased speech matches the audio). We also require the model to output the explanation and confidence level for its judgment to provide more reliable detection.  \
> Furthermore, the rejection rate is **34.77%**, which demonstrates the effectiveness and necessity of hallucination detection.

---

> ### Author Response · Authors · 2025-11-23
>
> ### W2: $R^3$-Bench-DX appears saturated as top models reach near-human accuracy, limiting its long-term utility.
> We respectfully argue that **high accuracy on individual questions does not imply that the benchmark is saturated or that the models have mastered the task.**
> 1. The "High Accuracy, Low Consistency" Paradox: A core contribution of our work is proving that current models are still far from solving the proposed social reasoning task. As shown in Table 3, there is a massive gap between Overall Accuracy and Chain Consistency ($Cons^c$):
>     - **GPT-4o**: Achieves **82.64%** accuracy but only **25.36%** in Chain Consistency.
>     - **Gemini 2.5 Pro**: Achieves **86.34%** accuracy but only **36.60%** in Chain Consistency.
> 2. Deconstructing Accuracy: "Hacking" vs. Understanding: To explain why this gap exists and why the benchmark is not saturated, we refer to the "Case 1 vs. Case 2" analysis (detailed in our response to Q3).  \
> **Case 1 (The "Hacking" Phenomenon)**: Our consistency analysis reveals that high-performing models frequently fall into "Case 1" scenarios—where they answer high-level causal questions correctly but fail on the preceding, simpler event/mental state questions. \
> This indicates that the "high accuracy" observed is often a result of models using heuristics or shortcuts (i.e., "hacking" the questions) rather than genuine causal reasoning. Unlike existing datasets where this issue goes detecting, $R^3$-Bench exposes that models have not truly "solved" the task. They are merely hitting the target without understanding the path. \
> Therefore, the benchmark remains highly challenging because solving it requires closing this consistency gap and eliminating "hacking" behaviors, which no current model has achieved.
>
> ### W3: The "Hard" subset (316 questions) is too small to serve as a robust evaluation set.
> We would like to clarify the design philosophy of our benchmark: $R^3$-Bench-DX and $R^3$-Bench-Hard are designed to be **complementary**, forming a holistic evaluation system where "Hard" serves as a specific high-difficulty probe, while "DX" provides large-scale diagnostic capability. \
> **DX as a Large-Scale Diagnostic Testbed**: $R^3$-Bench-DX contains **4,840 QA pairs**, which is statistically robust. Unlike traditional QA sets, DX is built upon **fine-grained mental-physical causal chains** (broken down into Event, Belief, Intent, Desire, Emotion)This structure allows us to perform deep "diagnostics" on why models fail. For example, in **Figure 5a**, we utilize the DX set to analyze model performance across six specific cognitive dimensions (e.g., "Contradiction in Words vs Behavior"), revealing fundamental deficits in multimodal conflict resolution. \
> **Hard as a Targeted Challenge**: The $R^3$-Bench-Hard set (316 QAs) is directly sourced from the winning submissions of a social reasoning challenge. Its purpose is not to serve as a standalone large-scale training/validation set, but to act as a **"stress test"** (the tip of the iceberg) to confirm the performance bounds on the most difficult samples. \
> **Synergy**: The failures on the Hard set are explained by the granular failures observed in the DX set. The DX set's consistency metrics ($Cons^c$ and $Cons^{sc}$) explain the reasoning disconnects that lead to lower performance on the Hard set. \
> Therefore, the robustness of our evaluation comes from the **combination** of the large-scale, structurally annotated DX set (for consistency and fine-grained diagnosis) and the Hard set (for peak difficulty assessment), rather than relying on the size of the Hard set alone.

---

> ### Author Response · Authors · 2025-12-03
>
> ### Q2: Unreported Human Consistency.
> Thank you for pointing it out. We fully acknowledge that a human baseline for the consistency metric serves as a crucial reference point. \
> The correlations between QAs corresponding to the same chain might give rise to shortcuts. For example, the correct option for a causal QA might appear in the question for another Node QA. Humans might exploit this shortcut, leading to an overestimation of human consistency performance. In our initially designed human study, each subject was restricted to answering only one question within a chain. This strong constraint caused human consistency to be severely underestimated. Consequently, we did not report human consistency metrics.
> However, as you pointed out, human consistency performance is indeed an important reference for models. Therefore, we conducted an **improved human study to provide human consistency performance for model reference**. First, we divided all QAs corresponding to a single chain into multiple groups based on the following principles:
> - All Node QAs constitute one group.
> - For causal QAs: Within a group, the node corresponding to the "correct option" of each QA must be distinct from the nodes corresponding to the "question" and "correct option" of any other QAs in the same group.
> **These principles ensure that no shortcuts exist within the same group.**
>
> Next, the **setting for the improved study** is as follows: for a chain, a subject is only allowed to answer one group of QAs. This ensures that subjects cannot exploit shortcuts while minimizing the underestimation of human consistency performance as much as possible. Under this setting, **the number of subjects required per chain decreased by an average of 7.44 compared to the initial human study, significantly alleviating the underestimation**. \
> Under this setting, we used the same QAs to conduct the improved study, obtaining human consistency performance across various metrics as shown in the table below:
>  | Emotion | Belief| Intent| Desire | MSE Overall | Event | CW | CH/W | Overall | $Cons^{c}$ | $Cons^{sc}$ |
> | --- | --- | --- | --- | --- | --- | --- | --- | --- | --- | --- |
> | 93.55%| 92.50%| 92.31% | 100% | 92.98% |  93.91%  | 90.21% | 92.37% | 92.24% | 60.47% | 75.52% |
>
> With a chain consistency of 60.47% and a subchain consistency of 75.52%, human significantly outperforms models (including fine-tuned ones), whose best consistencies reach only 36.60% and 60.95% respectively. This demonstrates that SOTA LVLMs still lag considerably behind human-level consistent reasoning.

---

### Official Review · Reviewer_aZqY · 2025-10-30

**Soundness:** 2
**Presentation:** 3
**Contribution:** 3
**Rating:** 4
**Confidence:** 4

**Summary:**

This paper introduces R3-VQA, a benchmark (R3-Bench about 5K QAs on diverse videos) and training dataset (R3-FDT about 40k QAs on movie videos) for evaluating video-based social reasoning. The dataset annotates causal chains linking beliefs, intents, desires, and emotions to observable actions.  Evaluation of 10+ SOTA models shows that they achieve high accuracy on individual questions but fail to maintain consistency across related questions in the same causal chain. The authors propose chain consistency and subchain consistency metrics to measure this. Training Qwen2-VL-7B with GRPO on R3-FDT improves performance by +13% on R3-Bench-DX and +6% on external social reasoning datasets, suggesting that causal structure provides useful training signal for social reasoning.

**Strengths:**

* Systematic annotation of mental-physical causal chains in video with multiple mental state types; consistency metrics effectively expose gap  in SOTA models
* Chain consistency improves dramatically (5.48% → 19.63%) after training, demonstrating models learn coherent multi-step "social" reasoning
* Rigorous construction: expert-verified annotations across multiple stages; comprehensive evaluation on 10+ models with accuracy and consistency metrics
* Strong competitive results: 7B model beats Gemini 1.5 Pro on hard set and approaches GPT-4o on standard benchmark; +6% generalization to external datasets (Social-IQ 2.0, IntentQA) also convincing
* Scalable automated pipeline (ARGUS) creatively leverages movie metadata to generate 41k training QAs; a valuable dataset for the community

**Weaknesses:**

* Related work somewhat underdeveloped: Dismisses MMToM-QA (2024, same year), Hi-ToM, SimpleTOM, and CausalChaos in single sentences without evidence or direct comparison—makes contribution difficult to situate relative to this work.
* Causal theory incomplete: Treats emotions as generic mental states rather than motivational drivers; ignores System 1/2 distinction between reactive (emotion-driven) and deliberative (belief-driven) reasoning. Also missing some ToM citations. Overall, the causal chains are reasonable but simplistic compared to cognitive processing.
* Human validation weak: No inter-rater agreement metrics. Consensus among annotators is not clearly discussed. It would be great if all the annotator information is provided in the dataset because social/affective labels are naturally ambiguous. Consistency in annotation is somewhat misguided.
* Domain characterization insufficient: Cross-domain gap acknowledged (movies → YouTube) but not quantified. No breakdown of R3-Bench by video type and no evaluation on genuinely real-world social scenarios.

**Questions:**

* MMToM-QA (2024) also annotates mental states in video. How does causal chain depth compare quantitatively?
*  What is performance using standard SFT (not GRPO) on the same 13k training examples?
* Can you provide error analysis showing whether model failures correlate with System 1 (reactive/emotion-driven) vs. System 2 (deliberative/belief-driven) scenarios?
* Why are Cohen's kappa or Fleiss' kappa not reported for causal chain annotations? What was the disagreement rate between the two expert reviewers before consensus?
*  What % of R3-Bench videos are "real-life" vs. curated (ads/short films)? Does performance differ by video type?
* Can you provide examples of Causal-Why questions where multiple answers seem valid? How were conflicts resolved?

---

> ### Author Response · Authors · 2025-11-23
>
> Thank you very much for your valuable time and for the careful review of our work! We try our best to address all your concerns as follows.
>
> ### W1&Q1: Underdeveloped Related Work and Quantitative Comparison of Causal Chain Depth with MMToM-QA.
> Thank you very much for pointing this out. As you mentioned, works such as MMToM-QA, Hi-ToM, SimpleTOM, and CausalChaos are indeed valuable contributions to our domain. In the revised paper, we have expanded the discussion of these works in **the third paragraph of the Introduction** and in **Section A.2 (Related Work) (highlighted in blue)**. Additionally, **Table 1** demonstrates the advantages of our dataset compared to other video QA datasets. \
> Regarding the quantitative comparison with MMToM-QA: MMToM-QA is an important benchmark for evaluating a model's ability to infer the beliefs and intentions of people in indoor environments. Causal chains are not annotated since they are not its primary focus. However, for a fair comparison, we **engage experts to annotate causal chains underlying each QA sample in MMToM-QA**, following the definition used in our work (as stated in Section 2.1). Statistical analysis reveals that the average causal chain length (the length of the longest continuous subchain, i.e., the causal chain depth) in MMToM-QA is **1.71**. This is lower than the average chain length of **3.3** in $R^3$-Bench-DX, indicating that our dataset is better suited for evaluating deep causal reasoning. Furthermore, our dataset encompasses a wider variety of mental states (belief, emotion, intent, and desire), features a broader space of beliefs and intents (unlike MMToM-QA, which is limited to beliefs and intents about indoor objects), and focuses on complex reasoning within realistic social scenarios. \
> We illustrate the disparity in causal depth between MMToM-QA and our dataset with an example. In MMToM-QA, a question is: 'If Thomas has been trying to get a condiment bottle, which one of the following statements is more likely to be true? (a) Thomas thinks that the condiment bottle is inside the microwave. (b) Thomas thinks that the condiment bottle is not inside the microwave.' The ground truth answer is 'a'. **The causal chain underlying this QA** is: 'Because Thomas tries to get a condiment bottle and he believes that the condiment bottle is inside the microwave **(reasons)**, he is poised to open the microwave **(result)**.' The length of this causal chain is 1. In contrast, **the causal chains annotated in our dataset are deeper and more complex**. For instance, the causal chain shown in Figure 2(a) involves causal transitions between external events and various mental states, including the intentions, beliefs, and emotions of Penny and the man.
> ### W2&Q3: Incomplete Causal Theory and Error analysis based on System 1/System 2
> Although we treat the emotion as a kind of mental states, its role as **motivational driver** **can be effectively captured** by annotating it as a reason node within a subchain. The distinction between **System 1 (emotion-driven)** and **System 2 (belief-driven)** scenarios can also be modeled by causal chains. Specifically, in a System 1 scenario, an emotion serves as a reason node leading to a person's action (external event). Similarly, in a System 2 scenario, a belief serves as a reason node. Our experts fully considered this distinction during annotation, correctly labeling the social interaction process representing these different systems. In summary, the causal chain is an effective method for modeling the causal relationships and dynamics between mental states and external events during social interactions, making them highly suitable for describing and modeling social scenarios.  \
> Regarding Q3, we greatly appreciate the insightful perspective you mentioned. **Subchains where emotion is the unique mental state reason represent System 1 scenarios, while subchains where belief is the unique mental state reason represent System 2 scenarios**. We report the models' performance on these two scenarios in detail in **Table 4 of Section 4.4**. We can find a remarkably consistent pattern among current LVLMs: their failures in System 1 scenarios are substantially more severe (across all metrics) compared to System 2 scenarios. This phenomenon reveals a critical deficiency in current LVLMs: while they are adept at understanding deliberative behavior, they fail to comprehend reactive (emotion-driven) behavior, which is a key factor hindering their social reasoning capabilities. In other words, the models severely **lack the ability to "empathize"**. This points to a **potential flaw in the current learning paradigm of large models**: learning from vast amounts of internet text data may enable the acquisition of rational thinking, but **fails to foster genuine "empathy"—a uniquely human capability that is crucial for social interaction.**

---

> ### Author Response · Authors · 2025-11-23
>
> ### W3&Q4: Disagreement metrics
> Thanks for your profound insight. It actually presents a dilemma: if the social/affective label is unique, we somewhat lose its natural ambiguity (Option 1); if we fully preserve the ambiguity, the QAs become too simple to effectively evaluate model capabilities (Option 2). Option 1 is the prevailing practice in current datasets; SocialIQ-2.0, IntentQA, and MELD (emotion recognition) all utilize unique ground truth answers, and we have followed this approach. Furthermore, we implement two measures to strike a balance: \
> (i) We retain data that is most likely to be unambiguous and widely accepted by people from different backgrounds (culture, age, etc.). **Each causal chain is approved by a crowdsource worker of arbitrary background and two experts**. In our dataset, each causal chain is approved by three people: the participant who designed the QA and two experts for verification (one of whom annotated the causal chain). The participant submits the QA and the textual description of the reasoning process behind it, establishing the main content of the causal chain. After being verified by two experts, the filtered data is approved by all three people. During the causal chain annotation and verification stage, the two experts do not alter the essence of the explanation of the causal chain. Therefore, we can consider the annotated causal chains to be widely acceptable. \
> (ii) **QAs represent the final form of our dataset, and we guarantee that the generated QAs are unambiguous**. Although interpretations of a social signal/mental state may vary, given the specific question and options, the ground truth answer is definitive. This allows our dataset to effectively mitigate the impact of ambiguity and provide reliable evaluation. As described in the verification principle (iv) in Section 2.2 under "QA Generation & Verification": "only one correct answer among five options," experts **revise ambiguous options** during QA verification. \
> The focus of "causal chain annotation" is to "resolve disagreements and reach consensus," ensuring the causal chains are as widely accepted as possible. Our approach involves iterative negotiation between the annotating expert and the verifying expert until they converge to a consensus (the version considered most likely to be widely accepted). During this iterative negotiation, experts generate multiple revised versions of the causal chain and continuously verify the new chains. This method differs from "verification-only" approaches where experts merely verify a single version; our scheme is more intricate and effective. Because the version of the causal chain being verified changes during our process, and the number of negotiation rounds for each chain is not fixed, **Cohen's kappa and Fleiss' kappa are not applicable**. However, we can report that the **disagreement rate** of the verifying expert regarding the initially annotated causal chains was **26%**.

---

> ### Author Response · Authors · 2025-11-23
>
> ### W4&Q5: Domain characterization is insufficient regarding the cross-domain gap (movies $\to$ YouTube). Please provide a quantitative breakdown of $R^3$-Bench by video type (e.g., % of Real-life vs. Curated) and analyze performance differences across categories.
> We thank the reviewer for highlighting the importance of domain characterization. We have performed a granular breakdown of our video sources and a detailed comparative evaluation to explicitly address the cross-domain gap and performance differences. \
> **(1) Quantitative Breakdown: Curated vs. Real-life** \
> To address your question regarding the dataset composition, we classified the 312 videos in $R^3$-Bench-DX based on the nature of the social scenarios. The vast majority (~94%) consists of Curated Social Scenarios, encompassing TV Series, Movie Clips, Sitcoms, and Advertising. We explicitly prioritized these sources because curated narratives involve high-density social signals, such as subtle acting, concealed intentions, and multi-layered irony, serving as a crucial testbed for probing advanced reasoning capabilities. The remaining ~6% represents Real-life Social Scenarios, including real-world user-uploaded content and Reality TV. \
> **(2) Performance Analysis: A Task-Dependent Divergence**\
> Our detailed performance comparison (see Figure 8 in in Section A.3.2) reveals a nuanced, task-dependent divergence. The performance gap is not uniform; rather, it depends on the direction of reasoning required by the task:\
> **Observation 1**: Curated data facilitates Perception and Deductive Reasoning (EU & CH/W). Models generally perform better on Curated data for Event Understanding (EU) and Causal-How/What (CH/W).\
> **Explanation**: According to our dataset design, CH/W questions focus on deductive reasoning (inferring effects from causes). Both EU and CH/W follow a forward-looking cognitive process—observing what is happening and predicting how it unfolds. Since curated content possesses higher production quality and clearer narrative structures, models can track this forward physical narrative more effectively. In contrast, Real-life videos contain a significant amount of irrelevant noise and distracting information. As the narratives in these scenarios are not pre-designed, they lack the clarity of curated content, resulting in lower performance in tracking event progression.\
> **Observation 2**: Curated data challenges Abductive Reasoning and Deep Cognition (CW & MSE). Crucially, this trend reverses for backward-looking tasks. For Causal-Why (CW) (abductive reasoning, inferring causes from effects) and Mental State Estimation (MSE), SOTA models often perform better on Real-life data than on Curated data.\
> **Explanation**: This suggests that the primary difficulty in Curated scenarios lies in their semantic depth. Unlike "Real-life" clips (often Reality TV) which tend to feature overt, exaggerated emotional conflicts, Curated narratives involve high-density social signals, such as subtle acting, concealed intentions, and multi-layered irony. These require the model to "read between the lines" rather than simply recognizing surface-level patterns.\
> **Consistency Analysis ($Cons^c$)**: This logical complexity creates a "reasoning bottleneck." As shown in Figure X (middle columns), the majority of evaluated models—including leading models like GPT-4o—exhibit lower reasoning consistency on Curated data compared to Real-life data. Despite observing physical events more accurately in Curated videos (High EU), these models frequently fail to maintain the logical chain (linking perception to intent) when facing the deeper narrative structures. This "High Perception, Low Consistency" phenomenon validates that our Curated subset effectively probes the upper limits of logical coherence.\
> **(3) Clarification on Cross-Domain Gap**\
> Regarding the transfer from Movie-based training to YouTube-based testing, we acknowledge a clear visual and production-style gap, but emphasize the logical alignment that bridges it:\
> **Visual & Production Divergence**: The domain gap is primarily defined by the shift in visual distribution. Our training set consists of high-fidelity, cinematic footage. In contrast, the test set (sourced from YouTube) introduces **diverse and uncontrolled visual conditions**, including compression artifacts, watermarks, hardcoded subtitles, and variable aspect ratios (e.g., vertical crops).\
> **Logical & Structural Alignment**: Despite these visual disparities, both domains share a unified **"Scripted Narrative Structure."** By training on movies, our model learns the abstract Social Syntax—the causal rules governing how beliefs and desires drive actions.\
> The robust performance on the test set demonstrates that this underlying social logic is invariant to production style. Our GRPO fine-tuning successfully equips the model with these **pure, abstract social reasoning capabilities**, allowing it to generalize effectively across the visual domain gap.

---

> ### Author Response · Authors · 2025-11-23
>
> ### Q6: Examples for answer ambiguity and conflict resolution in Causal-Why questions.
>
> We provide two cases where multiple answers seem valid and the reasonable options other than the answers have been modified by expert annotators. \
> In the first case, the man named Mikael was told to pump at 100 times per minute. He then asked how many times he should pump per hour and said he could divide and count to it. The man in the white shirt was puzzled by this. The question in our dataset reads as follows: "Why does the man in the white shirt feel confused?" The correct answer is: "He feels confused because Mikael's approach to calculating beats per minute seems redundant." There is another option in the original list that also makes sense: "He feels confused because he doesn't understand why Mikael is asking about beats per hour when they are discussing beats per minute." The annotator has revised it to: "He feels confused because he fully understands why Mikael is asking about beats per hour when they are discussing beats per minute." This ensures that only the correct answer makes sense. \
> In the second case, the woman in green pretended to have heard of Claire's stories in order to show closeness, but when Claire asked about specific examples, the woman felt embarrassed and couldn't come up with an answer. The question is: "Why did the woman in green feel embarrassed after Claire asked for examples?" The correct answer is: "She felt embarrassed because she didn't actually know anything about Claire despite her earlier statement." One of the original options also makes sense: "She felt embarrassed because she had exaggerated her knowledge about Claire to impress others." The annotator has revised it to: "She felt embarrassed because she had revealed her knowledge about Claire to impress others." Overall, our annotators revised those options that were also reasonable, ensuring that only the correct answer was reasonable.

---

> ### Author Response · Authors · 2025-12-03
>
> ### Q2: What is performance using standard SFT (not GRPO) on the same 13k training examples?
> Per your suggestion, we conducted a standard Supervised Fine-Tuning (SFT) experiment using the exact same 13k training examples. The results are summarized in the table below, providing compelling evidence for the effectiveness of our dataset.
>
> | Dataset | Qwen2-VL-7B (Base) | **Ours-FT (SFT)** | **Ours-FT (RLFT)** |
> | :--- | :---: | :---: | :---: |
> | **SocialIQ-2.0** | 61.51 | 65.38 | **68.40** |
> | **IntentQA** | 82.38 | 87.77 | **90.58** |
> | **R$^3$-Bench-DX** | 74.90 | **88.74** | 86.88 |
> | **R$^3$-Bench-Hard** | 31.96 | **42.09** | 39.87 |
>
> First, the data proves highly effective regardless of the training method. Both **Ours-FT (SFT)** and **Ours-FT (RLFT)** yield significant improvements over the Qwen2-VL-7B baseline across all four benchmarks. This consistency confirms that the performance gains reported in our paper are primarily driven by the high quality and information density of the proposed dataset itself, rather than being solely dependent on the specific RL algorithm.
>
> When comparing the two methods, we observe that SFT is particularly efficient at fitting the structural priors of the data. Since *R$^3$-Bench-DX* and *Hard* share a similar causal chain structure with our training set, **Ours-FT (SFT)** actually achieves slightly higher performance on these benchmarks, indicating that SFT is excellent at capturing the specific reasoning patterns present in the training distribution.
>
> On the other hand, **Ours-FT (RLFT)** demonstrates broader generalization capabilities, achieving higher scores on *SocialIQ-2.0* and *IntentQA*. This suggests that while SFT is superior for exploiting similar structures, the reinforcement learning objective encourages the model to generalize better.
>
> Regarding the training setup, we converted 10% of the training samples into an open-ended QA format (i.e., removing the options and requiring direct answers). We did this simply to prevent catastrophic forgetting; without this mixture, the model tends to "overfit" to the multiple-choice format and loses the ability to generate natural responses. Our RLFT method (using GRPO) does not require this step because its exploration mechanism naturally preserves generation capabilities.

---

### Official Review · Reviewer_DYRn · 2025-11-01

**Soundness:** 3
**Presentation:** 3
**Contribution:** 3
**Rating:** 6
**Confidence:** 3

**Summary:**

The paper introduces R3-VQA, a new video social-reasoning suite comprising (i) R3-Bench—an evaluation benchmark with fine-grained mental–physical causal chains and (ii) R3-FDT—a large-scale training set generated by an automated pipeline. The benchmark defines four QA types—Event Understanding, Mental State Estimation, Causal-Why, and Causal-How/What—generated from expert-annotated chains and verified by the domain experts. Reported scale: around 0.3k videos / 5k MC-QAs for R3-Bench and, for R3-FDT, about 3k videos / 68k MC-QAs (Table 1); elsewhere, the text states 2,812 videos / 41k QAs after quality filtering as well. Evaluations of many LVLMs demonstrate a stark gap between single-question accuracy and chain/subchain consistency (all QAs from a chain must be not wrong), highlighting fragmented reasoning. Fine-tuning Qwen2-VL-7B with GRPO on R3-FDT yields sizable gains on R3-Bench and transfers to Social-IQ 2.0 dataset / IntentQA dataset.

**Strengths:**

1. Explicit belief/intent/desire/emotion nodes and chain/subchain QA generation tie items to causal structure rather than one-off facts.
2. Two-stage expert checks for QAs and chains; a clear human-in-the-loop pipeline to keep the ambiguity low.
3. Chain and Subchain Consistency expose fragmented reasoning hidden by the average accuracy; formal definitions are offered.
4. Large consistency-accuracy gaps ( for example, GPT-4o 82.64% overall vs 25.36% chain consistency; Gemini 2.5 Pro 86.34% vs 36.60%) and dimension-wise weaknesses.
5. Automated R3-FDT pipeline (alignment, movie scripts, Gemini filtering, GPT-4o) plus GRPO improves Qwen2-VL-7B model ( for instance, plus 32.00% performance on R3-Bench-DX; and plus 9.81% on R3-Bench-Hard).

**Weaknesses:**

1. The OE scoring depends on an LLM judge. What is more, the sensitivity to prompts and judgment choice is not reported comprehensively. Frame or frame per second budgets differ comparatively over models. This confounds the absolute rankings to a considerable extent.
2. 3k/68k for R3-FDT is stated in Table 1; however, the text reports 2,812/41k later. It is better to clarify the final counts and which subset is utilized for the GRPO algorithm to make the paper clearer.
3. R3-FDT depends on movies and LLM-generated chains/QAs. The risk of stylistic bias or leakage still remains; in addition, the human IAA and leakage audits are not detailed.
4. Even though consistency is measured, wall-clock latency and user-utility vs delay are not evaluated directly.
5. The human chain or the subchain consistency is left out (with justification deferred to the Appendix), which limits context for the new metrics.

**Questions:**

1. How do OE and consistency scores change if the judge-LLM or prompt is varied? Any agreement with human raters on an OE slice?

---

> ### Author Response · Authors · 2025-11-23
>
> Thanks for your insightful suggestions on our work and valuable time! We address the concerns below.
> ### W1&Q1 OE and consistency scores, experimental settings (e.g., frame count, FPS).
> We apologize for the confusion. In fact, in the evaluation experiments on $R^3$-Bench-Hard and $R^3$-Bench-DX, **we did not employ any LLM judge to score QA accuracy or consistency**. We utilized VLMEvalKit, a toolkit widely accepted and recognized by the community, to evaluate SOTA LVLMs. **We adopted its rule-based scoring strategy (specifically, exact matching)** to ensure that our evaluation settings are aligned with community standards. Consequently, our evaluation experiments do not involve the sensitivity to prompts and judgment choice, nor do they concern the agreement with human raters. \
> Regarding the settings such as frame budgets, which you noted could confound the absolute rankings, we kept these **consistent across all models** to ensure a fair ranking. Specifically, we provided a visual context of **16 frames at a 640x360 resolution to all models**. Since the Gemini family of models supports native video inputs, for Gemini 1.5 Pro and Gemini 1.5 Flash, we additionally reported evaluation results using video inputs. As the performance of these two models under the video input setting (denoted as Gemini 1.5 Flash (Video) and Gemini 1.5 Pro (Video) in Tables 2, 3, and 8) was inferior to their performance under the frame input setting (denoted as Gemini 1.5 Flash (Frames) and Gemini 1.5 Pro (Frames)), we did not report video input results for Gemini 2.5 Pro. In summary, our experimental setup ensures a fair ranking. We have added details of our experimental setup at the beginning of 'Section 4.1 Evaluation' (highlighted in blue) to clarify it.
> ### W2: Unclear count for $R^3$-FDT.
> The final counts for R3-FDT are indeed 2812 videos/41k QAs. The subset utilized for the GRPO algorithm training consists of 13k QAs that were randomly selected from 41k QAs. The initial 3k/68k figures stated in Table 1 represented the counts before our data correction and hallucination detection process. We inadvertently overlooked correcting this entry in the table during the initial submission. We have now corrected tit in the revised paper and have double-checked to ensure that all dataset statistics are accurate and consistent throughout the manuscript to make the paper clearer.
> ### W3: For $R^3$-FDT: the risk of stylistic bias or leakage and incomplete description of the human IAA and leakage audits.
> $R^3$-FDT is a large-scale training dataset generated by a novel, automated pipeline, designed to facilitate the development of foundation models in this domain. As stated at the beginning of Section 3, this pipeline was proposed to **overcome the two major limitations** constraining the development of foundation datasets and foundation models: data scarcity and the high cost of manual annotation. Therefore, $R^3$-FDT **does not involve any manual effort** and thus does not require human IAA and leakage audits. \
> Although we have strived to enhance the quality of $R^3$-FDT through multiple rounds of model validation, **we acknowledge that a certain risk of stylistic bias or leakage may still exist**. However, **for a training dataset, we believe it is acceptable**. The performance improvement and **generalization ability** of the model shown in Figure 5(b), after being fine-tuned on $R^3$-FDT, **serve as the best evidence of its effectiveness**.
> ### W4: Utility vs delay/latency.
> Thank you very much for your question. Could you please clarify the specific meanings of 'user-utility' (accuracy? consistency? or both?) and 'delay/wall-clock latency' (time for answering? cost?) mentioned in your review? We sincerely look forward to your further clarification, and we will conduct supplementary experiments to try to fully address your concerns.

---

> ### Author Response · Authors · 2025-12-03
>
> ### W5: Unreported Human Consistency.
> The correlations between QAs corresponding to the same chain might give rise to shortcuts. For example, the correct option for a causal QA might appear in the question for another Node QA. Humans might exploit this shortcut, leading to an overestimation of human consistency performance. In our initially designed human study, each subject was restricted to answering only one question within a chain. This strong constraint caused human consistency to be severely underestimated. Consequently, we did not report human consistency metrics.
> However, as you pointed out, human consistency performance is indeed an important reference for models. Therefore, we conducted an **improved human study to provide human consistency performance for model reference**. First, we divided all QAs corresponding to a single chain into multiple groups based on the following principles:
> - All Node QAs constitute one group.
> - For causal QAs: Within a group, the node corresponding to the "correct option" of each QA must be distinct from the nodes corresponding to the "question" and "correct option" of any other QAs in the same group.
> **These principles ensure that no shortcuts exist within the same group.**
>
> Next, the **setting for the improved study** is as follows: for a chain, a subject is only allowed to answer one group of QAs. This ensures that subjects cannot exploit shortcuts while minimizing the underestimation of human consistency performance as much as possible. Under this setting, **the number of subjects required per chain decreased by an average of 7.44 compared to the initial human study, significantly alleviating the underestimation**. \
> Under this setting, we used the same QAs to conduct the improved study, obtaining human consistency performance across various metrics as shown in the table below:
>  | Emotion | Belief| Intent| Desire | MSE Overall | Event | CW | CH/W | Overall | $Cons^{c}$ | $Cons^{sc}$ |
> | --- | --- | --- | --- | --- | --- | --- | --- | --- | --- | --- |
> | 93.55%| 92.50%| 92.31% | 100% | 92.98% |  93.91%  | 90.21% | 92.37% | 92.24% | 60.47% | 75.52% |
>
> With a chain consistency of 60.47% and a subchain consistency of 75.52%, human significantly outperforms models (including fine-tuned ones), whose best consistencies reach only 36.60% and 60.95% respectively. This demonstrates that SOTA LVLMs still lag considerably behind human-level consistent reasoning.

---

### Official Review · Reviewer_curi · 2025-11-05

**Soundness:** 3
**Presentation:** 3
**Contribution:** 3
**Rating:** 6
**Confidence:** 3

**Summary:**

The paper introduces the video social reasoning task with fine-grained mental–physical causal chains. The paper releases an evaluation benchmark with verified Multiple choice-QA pairs built from annotated chains and a training set generated via a movie-script alignment and LLM pipeline. The paper introduces chain and subchain consistency metrics, evaluate many LVLMs and humans, and show GRPO fine-tuning of Qwen2-VL-7B improves accuracy and consistency.

**Strengths:**

- Task novelty: Causal-chain–grounded evaluation of belief, intent, desire, and emotion in video; chain and subchain consistency move beyond single-QA accuracy.
- Dataset annotation quality: Human-verified chains and QA, with explicit rules and hallucination filtering
- Significance: Reveals large accuracy-consistency gaps in SOTA LVLMs and provides training data that measurably improves a strong open model.
- Strong experiment results with ablation studies and generalization tests.

**Weaknesses:**

- Human study mismatch. Humans answer only one QA per chain, so their consistency is “severely underestimated,” complicating human–model comparisons. Remedy: redesign human protocol to mirror model setting or normalize metrics accordingly.
- LLM-in-the-loop biases/leakage. GPT-4o is used to generate and self-correct chains and QAs; Gemini is used for hallucination detection. This can imprint model priors and evaluation artifacts. It'll be good to do ablation studies to assess its sensitivity to the design choice.
- Lack of uncertainty estimation. Many tables lack confidence intervals and seed variance.

**Questions:**

- how does the performance correlate with the chain length?

---

> ### Author Response · Authors · 2025-11-23
>
> Thanks so much for taking the time to review our work and giving valuable feedback! We address all your concerns below.
>
> ### W2: Assess the sensitivity to the design choice
> The LLM-in-the-loop pipeline **is utilized exclusively for generating our training set, $R^3$-FDT, and has no impact on our evaluation set, $R^3$-Bench**. To assess the sensitivity to design choices, we evaluate the **consistency** between the results of other models and the models we actually employed (GPT-4o for redundant chain removal in self-correction and Gemini 2.5 Flash for hallucination detection). In both of these two stages, the model decides whether to keep or remove a chain/QA. Therefore, consistency can be defined as the proportion of choices where the two models agree. \
> For **self-correction**, the consistencies between the choices made by other models and GPT-4o are shown in the table below:
> | Gemini 2.5 Flash | Gemini 2.0 Flash | GPT-5|
> | --- | --- | --- |
> | 0.8534 | 0.8931 | 0.8593 |
>
> For **hallucination detection**, the consistencies between the choices made by other models and Gemini 2.5 Flash are presented in the following table:
>  | GPT-5 Mini | Gemini 2.0 Flash| GPT-4.1 Mini|
> | --- | --- | --- |
> | 0.8782 | 0.9625 | 0.9845 |
>
> The results demonstrate a high degree of consistency in the choices made by different models across both stages, indicating that the results are **not sensitive** to different design choices. However, we acknowledge that there are certain differences among models. We will **release the selection results and selected chains&QAs from different models upon acceptance to facilitate flexible usage**.
>
> ### W3: Lack of uncertainty estimation. Many tables lack confidence intervals and seed variance.
> For video question answering tasks, deterministic evaluation is the standard and widely adopted practice. **The results reported by leading models such as Gemini and GPT are also deterministic**, without providing uncertainty estimations like confidence intervals or seed variance.
> For instance, this can be observed in **Table 6** of the Gemini 2.5 report (https://arxiv.org/pdf/2507.06261) and in the **"Video long context"** subsection of the **"Vision"** section in the GPT-4.1 report (https://openai.com/index/gpt-4-1/).
> In line with this standard, we employed the widely **accepted evaluation toolkit, VLMEvalKit**, which also performs deterministic evaluation. Furthermore, to **ensure our results are reproducible**, we set the temperature to 0 during all evaluations.
>
> ### Q1: How does the performance correlate with the chain length?
> Thank you for providing this excellent perspective for analysis. We have supplemented our paper with additional experimental results (**Figure 6**) and analysis in Section 4.4 (under the paragraph "The correlation between performance and chain length").  \
> In practice, **the causal chain is used to model the interrelationship between a character's mental state and external events during social interactions**. A longer chain naturally implies a greater number of transitions between these internal states and external events. As illustrated in Figure 6, we observe that the difficulty of atomic units—specifically **a single event (EU QA)**, **a mental state (MSE QA)**, or **an atomic interaction process (Subchain Consistency, $Cons^{sc}$)**—is **independent** of the chain length. Consequently, these metrics remain stable and do not exhibit significant fluctuations as the chain extends.  \
> **However**, a clear negative correlation is observed in the global consistency metric **($Cons^{c}$)**. This highlights a critical insight: while models maintain robust performance on **local** reasoning steps, the errors caused by inconsistent reasoning **accumulate and propagate** throughout the complex web of interrelationships as the chain lengthens. Thus, the decline in $Cons^{c}$ reflects the model's vulnerability to **cascading errors** in long-horizon causal modeling, rather than a lack of understanding of individual social interactions.

---

> ### Author Response · Authors · 2025-12-03
>
> ### W1: Human study mismatch.
> We sincerely appreciate your suggestion, which is crucial for facilitating valid human–model comparisons. \
> The correlations between QAs corresponding to the same chain might give rise to shortcuts. For example, the correct option for a causal QA might appear in the question for another Node QA. Humans might exploit this shortcut, leading to an overestimation of human consistency performance. In our initially designed human study, each subject was restricted to answering only one question within a chain. This strong constraint caused human consistency to be severely underestimated. Consequently, we did not report human consistency metrics.
> However, as you pointed out, human consistency performance is indeed an important reference for models. Therefore, we conducted an **improved human study to provide human consistency performance for model reference**. First, we divided all QAs corresponding to a single chain into multiple groups based on the following principles:
> - All Node QAs constitute one group.
> - For causal QAs: Within a group, the node corresponding to the "correct option" of each QA must be distinct from the nodes corresponding to the "question" and "correct option" of any other QAs in the same group.
> **These principles ensure that no shortcuts exist within the same group.**
>
> Next, the **setting for the improved study** is as follows: for a chain, a subject is only allowed to answer one group of QAs. This ensures that subjects cannot exploit shortcuts while minimizing the underestimation of human consistency performance as much as possible. Under this setting, **the number of subjects required per chain decreased by an average of 7.44 compared to the initial human study, significantly alleviating the underestimation**. \
> Under this setting, we used the same QAs to conduct the improved study, obtaining human consistency performance across various metrics as shown in the table below:
>  | Emotion | Belief| Intent| Desire | MSE Overall | Event | CW | CH/W | Overall | $Cons^{c}$ | $Cons^{sc}$ |
> | --- | --- | --- | --- | --- | --- | --- | --- | --- | --- | --- |
> | 93.55%| 92.50%| 92.31% | 100% | 92.98% |  93.91%  | 90.21% | 92.37% | 92.24% | 60.47% | 75.52% |
>
> With a chain consistency of 60.47% and a subchain consistency of 75.52%, human significantly outperforms models (including fine-tuned ones), whose best consistencies reach only 36.60% and 60.95% respectively. This demonstrates that SOTA LVLMs still lag considerably behind human-level consistent reasoning.

---

### Author Response · Authors · 2025-12-03
**Summary of Work and Rebuttal**

Dear Area Chair,

We would like to express our sincere gratitude to you and all reviewers for your hard work and careful review. Your valuable suggestions have been instrumental in helping us further refine our work. Here, we provide a brief summary of our work and the rebuttal.

In this work, we model the causal relationships between characters' mental states and external events in complex social scenarios using **causal chains**. Based on this, we propose a high-quality evaluation benchmark, $R^3$-Bench, and a large-scale training dataset, $R^3$-FDT, to address current limitations in the domain of video social reasoning. $R^3$-Bench contains causal chains and QAs meticulously annotated and verified by experts, further divided into a challenging $R^3$-Bench-Hard and a diagnostic $R^3$-Bench-DX. $R^3$-FDT is generated by a novel automated pipeline and similarly contains causal chains and QAs to support the training of foundation models. Extensive experiments demonstrate that: (i) SOTA LVLMs fail at complex social reasoning tasks, exhibit significant inconsistency, and remain far from human-level performance; and (ii) $R^3$-FDT can significantly enhance models' video social reasoning capabilities and generalization.

We are encouraged that the reviewers unanimously praised the paper's novelty and depth of evaluation. Specifically, Reviewers **curi, DYRn, and aZqY** highlighted that the introduction of consistency metrics based on causal chains (beliefs/intentions/emotions) effectively reveals the reasoning gaps hidden beneath the high accuracy scores of SOTA models, while Reviewer **G5Ui** also noted the significant performance gap between AI and humans on the hard subset. The dataset construction quality was highly commended. Reviewers **curi, DYRn, and aZqY** praised the multi-stage expert verification and strict annotation guidelines that ensure high reliability and low ambiguity, with Reviewer **G5Ui** further acknowledging the dataset's rich annotation and decent scale. Furthermore, the automated training pipeline ($R^3$-FDT) and experimental results were found to be compelling, demonstrating not only significant improvements in the consistency and generalization of open-source models (surpassing some closed-source models) but also the scalability and effectiveness of the pipeline (Reviewers **curi, DYRn, aZqY, G5Ui**).

Overall, the rebuttal includes the following contents:
* **Fine-grained Analysis (mentioned by Reviewer curi, Reviewer aZqY, and Reviewer G5Ui)**. We find that the causal chain is a framework highly suitable for fine-grained analysis and providing valuable insights. Leveraging them, we analyzed the differences in model performance across System 1 and System 2 scenarios. We discovered that models actually perform worse on System 1 scenarios, pointing directly to a potential flaw in the current learning paradigm of large models: it is difficult to truly learn the crucial capability of "empathy" merely from vast amounts of text on the internet. We also found that the proposed consistency metric can accurately detect the hijacking phenomenon in causal questions and provide a more reliable evaluation. We analyzed the relationship between model performance and chain length, finding that errors caused by models' inconsistent behavior accumulate as the chain length grows. Through the rebuttal, we realized that causal chains offer more perspectives and possibilities for fine-grained analysis, which is essential for uncovering the nature of model defects.
* **Improved Human Study (mentioned by Reviewer curi, Reviewer DYRn, and Reviewer G5Ui)**. We conducted an improved human study to provide a reasonable human consistency performance for model reference. The results indicate that there is still a significant gap between the reasoning consistency of current SOTA LVLMs (best at 36.60% and 58.82%) and human levels (60.47% and 75.52%).
* **Supervised Fine-Tuning (SFT) Experiments (mentioned by Reviewer aZqY)**. We supplemented experiments on SFT using $R^3$-FDT. The results show that models after RLFT (reported in original paper) and SFT achieve significant performance improvements on $R^3$-Bench and two other related datasets (IntentQA and SocialIQ 2.0), fully demonstrating the value of $R^3$-FDT.
* **Other Concerns**. We have also actively responded to other concerns raised by reviewers. Through supplementary experiments and clarifications, we believe these concerns have been well addressed.

We highlighted all modifications in blue in the revised paper and indicated locations of references in the comments. We hope this facilitates your review.

In conclusion, we believe we have comprehensively addressed the reviewers' concerns and further verified the value of our work through supplementary experiments. We thank you and all reviewers once again for your valuable feedback. We firmly believe that this work will make a significant contribution to the domain of video social reasoning.

Best regards.

---

### Meta-Review · Area_Chair_C5bQ · 2025-12-17

**Summary:**

This paper presents a new benchmark and a new dataset to address the challenge of teaching AI to infer complex social and mental states from videos. The benchmark contains four QA types, including Event Understanding, Mental State Estimation, Causal-Why, and Causal-How/What, which are generated from expert-annotated chains and verified by the domain experts. In particular, it contains fine-grained annotations of belief, intent, desire, emotion, and their causal chains in complex scenarios. The dataset is generated through a new automated pipeline with the same structure. The authors conducted comprehensive evaluations of state-of-the-art large vision-language models (LVLMs) on the benchmark and also finetuned a 7B model using the new dataset. Extensive discussions and insights are provided in the paper.

Reviewers agreed that this paper develops a novel benchmark and provides very comprehensive evaluations. In particular, the introduction of consistency metrics based on causal chains (beliefs/intentions/emotions) could potentially reveal the reasoning gaps in SOTA models. This work presents a valuable benchmark and dataset for the community.

**Reviewer Concerns:**

Reviewers also raised concerns regarding fine-grained analysis of the evaluations, supervised fine-tuning experiments, evaluation metrics, human study mismatch, missing related work,  as well as some minor issues (e.g., unclear statistics of dataset). The authors have provided detailed responses with additional experimental results in their rebuttal, which have addressed most of the major concerns from reviewers (based on my understanding). A few minor issues such as latency and causal theory shall be further clarified in the final version.

**Reviewer Scores:**

This paper received mixed initial ratings (6, 6, 6, 4). I think major concerns from the reviewer who gave a rating of 4 have been well addressed in the rebuttal.

---

### Decision · Program_Chairs · 2026-01-26

Accept (Poster)